# Grik2b and Grik2c kainate receptors regulate oviposition in *Bactrocera dorsalis*

**Bin Liu[1], Jingwei Yang[1], Long Ye[1], Yang Xiao[1], Guohong Luo[1], Muyang He[2], Guy Smagghe[3,4,5], Yongyue Lu[1], Daifeng Cheng**◉[1]*

1 Department of Entomology, South China Agricultural University, Guangzhou, China, 2 Zhongkai University of Agriculture and Engineering, Guangzhou, China, 3 Department of Plants and Crops, Ghent University, Ghent, Belgium, 4 Institute of Entomology, Guizhou University, Guiyang, China, 5 Molecular and Cellular Life Sciences, Department of Biology, Vrije Universiteit Brussel (VUB), Brussels, Belgium

* chengdaifeng@scau.edu.cn

## Abstract

Oviposition holds crucial significance for insect reproduction. Nevertheless, the research on the neural conduction mechanism of oviposition is still rather limited in most agricultural pests. Here, we demonstrate that the conserved Kainate receptors (KARs) expressed in the glutamatergic neurons (GNs) and the ovipositor neuromuscular junctions (NMJs) regulate the oviposition behavior in *Bactrocera dorsalis*. We identified two KARs (Grik2b and Grik2c), which control the oviposition behavior by influencing both oviposition preference and egg-laying quantity. Protein-ligand interaction indicated that glutamate serves as the neurotransmitter of Grik2b and Grik2c. Knockdown glutamate-coding genes adversely impacted oviposition preference and egg-laying quantity. Specific knockdown Grik2b (or Grik2c) in the GNs and NMJs could respectively influence oviposition preference and egg-laying quantity. Finally, inhibitors of KARs were screened for their ability to inhibit oviposition. Our study provides strong supporting evidence that a novel neural conduction mechanism for oviposition by uncovering the diverse roles of KARs and provides potential molecular target controlling insect oviposition.

## Introduction

Oviposition plays a crucial role in the generation of offspring and the sustainability and prosperity of insect populations and species. A comprehensive understanding of the oviposition mechanisms of female insects is essential for species propagation and pest control [1,2]. The coordination between the peripheral sensory system and the central nervous system (CNS) is necessary for executing the process of depositing eggs [3]. Multiple sensory systems and receptors have been confirmed to guide oviposition in insects [4–6]. However, study about internal neurons and circuits involved in making oviposition decision is limited to *Drosophila* [7,8] and various

**Data availability statement:** All the data needed to evaluate the conclusions in the paper are presented in the paper and/or the Supplementary Materials. RNA-sequencing data have been deposited in the Genome Sequence Read Archive Database of the National Genomics Data Center (PRJCA004790). Gene accesssion numbers of the genes in this study are XP049313481, XP011207794, XP011210541, XP029408769, XP049318226, XP011202215, XP011208683, XP029409101, XP011198795 and XP049307715.

**Funding:** This study was supported by the National Key Research and Development Program of China (https://service.most.gov.cn/) (No. 2025YFC2609002 to DC, 2023YFD1401401 to DC), the National Natural Science Foundation of China (https://www.nsfc.gov.cn/) (No. 32372520 to DC, No. 32122072 to DC), Guangdong Special Support Program-Leading Talents in Science and Technology Innovation (https://gdstc.gd.gov.cn/) (2023TX07A017 to YL) and Science and Technology Talent and Platform Project of Yunnan (Academician and Expert Workstation) (https://kjt.yn.gov.cn/) (202405AF140082 to YL). The funders had no role in study design, data collection and analysis, decision to publish, or preparation of the manuscript.

**Competing interests:** The authors have declared that no competing interests exist.

**Abbreviations:** CNS, central nervous system; D-AP5, D-2-Amino-5-phosphonovaleric acid; DEGs, differentially expressed genes; EC, sethyl caprylate; FDR, false discovery rate; FISH, Fluorescence In Situ Hybridization; GluDH, glutamate dehydrogenase; GluSN, glutamate synthase; GO, Gene Ontology; 3-HA, 3-hexenyl acetate; HE, hematoxylin and eosin; iGluRs, ionotropic glutamate receptors; KARs, kainate receptors; LBDs, ligand-binding domains; ML, maximum likelihood; MST, microscale thermophoresis; NBQX, 2, 3-dihydroxy-6-nitro-7-sulfamoyl-benzo (F) quinoxaline; NMJ, neuromuscular junction; qRT-PCR, quantitative real-time PCR; SPR, surface plasmon resonance; TEM, transmission electron microscopy; Vglut, vesicular glutamate transporter.

receptors are involved in the communication between neurons and their target organs in a neurotransmitter-dependent manner [9–12].

While the CNS plays a crucial role as the decision-making center for oviposition, it is important not to overlook the role of the ovipositor. Research has revealed that odorant receptors are present in the ovipositors of various insect species [13–16]. Although there is a certain level of understanding regarding peripheral receptor involvement in odorant recognition within the ovipositor, the internal neurons related to oviposition remain elusive.

In *Drosophila*, glutamate has already been implicated in driving the egg-laying motor program via the internal sensory neurons and abdominal ganglion neurons [17,18]. Glutamate serves as the principal excitatory neurotransmitter within the CNS. Its actions are predominantly mediated through three classes of ionotropic glutamate receptors (iGluRs), namely AMPA receptors, kainate receptors (KARs), and NMDA receptors [19]. The subsequent cloning of insect iGluRs has disclosed sequence similarities with their vertebrate counterparts of AMPA, kainate, and NMDA receptors. Nevertheless, with the sole exceptions of the neuromuscular junction (NMJ) in larval *Drosophila* [20] and the NMJ of adult locusts [21], the small size and inaccessibility of insect neurons have to date challenged characterization of the functional properties of native insect iGluRs. In *Drosophila*, four putative KARs are functionally indispensable for spectral preference behavior [22]. The eye-enriched kainate receptor is expressed in photoreceptors, receiving feedback glutamatergic signals from amacrine cells [23]. However, to this point, in vitro reconstitution has not been accomplished for any of these putative KARs.

The oriental fruit fly, *Bactrocera dorsalis*, is a significant pest inflicting considerable damage to fruit and vegetable production. Typically, the females of *B. dorsalis* inflict damage upon the fruits by depositing the eggs deep within the fruits using their tapered ovipositor [24]. Studies have indicated that gravid females of *B. dorsalis* exhibit a pronounced preference for oviposition on host fruits containing gut bacteria (*Citrobacter* sp.) and specific volatiles (3-hexenyl acetate (3-HA) and ethyl caprylate (EC)) [25,26]. In this study, we have provided strong supporting evidence that the neural conduction mechanism underlying the behavioral alterations in oviposition of *B. dorsalis* by identifying two receptors expressed in the glutamatergic neurons (GNs) and ovipositor NMJs.

## Results

### KARs were highly expressed in the ovipositor of gravid female

The oviposition-associated genes in the ovipositor was screened by transcriptome analysis of ovipositors at different developmental stages, ranging from newly emerged females (0-day-old, 3-day-old, 6-day-old and 9-day-old, unmated) to gravid females (12-day-old, mated). The results revealed significant differences in gene expression patterns among ovipositors at different developmental stages; with closer age proximity resulting in greater similarity to the gene expression pattern of gravid females (12-day-old) (Figs 1A, 1B, and S1). Furthermore, gene expression trend analysis generated 20 distinct gene expression profiles (S2 Fig), with notable

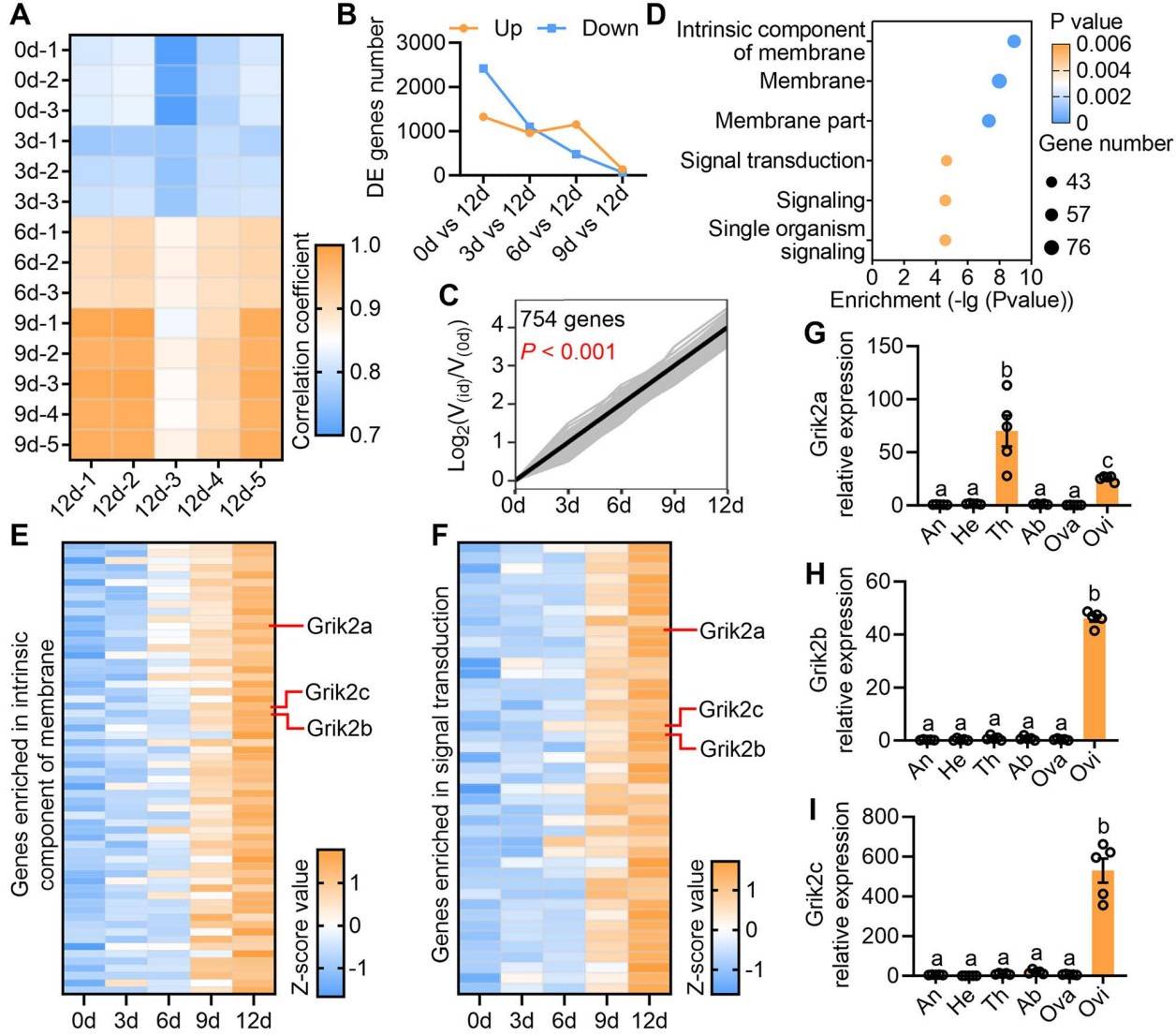

**Fig 1. KARs were highly expressed in the ovipositor of gravid female. (A)** Pearson correlation analysis shows differences between female ovipositor gene expression patterns at different developmental stages. -1, -2, -3, -4, and -5 represent different replicates of the sampling time (0d, 3d, 6d, 9d, and 12d). **(B)** The number of differentially expressed genes (DEGs) in the ovipositor of 12-day-old females compared with junior females. **(C)** The gene expression profile in which gene expression is higher in ovipositor of senior females. $V_{(0d)}$ means expression of genes in 0d sample, $V_{(id)}$ means expression of genes in 3d, 6d, 9d, and 12d samples. **(D)** Top 3 GO pathways (Cellular component and Biological process) enriched with 754 DEGs in **(C)**. **(E)** Expression pattern of DEGs enriched in intrinsic component of membrane pathway. **(F)** Expression pattern of DEGs enriched in signal transduction pathway. **(G)** Tissue expression of Grik2a ($n = 5$, $F_{(5,24)} = 22.11$, $P < 0.0001$, Ordinary one-way ANOVA). **(H)** Tissue expression of Grik2b ($n = 5$, $F_{(5,24)} = 1,111$, $P < 0.0001$, Ordinary one-way ANOVA). **(I)** Tissue expression of Grik2c ($n = 5$, $F_{(5,24)} = 72.74$, $P < 0.0001$, Ordinary one-way ANOVA). An: antenna; He: head; Th: thorax; Ab: abdomen; Ova: ovary; Ovi: ovipositor. The data underlying this figure can be found in S1 Data.

enrichment of 754 genes exhibiting higher expression levels in the ovipositors of senior females (Fig 1C). Subsequent Gene Ontology (GO) analysis indicated a significant enrichment of these genes in membrane components and terms related to signal transduction (Fig 1D). Notably within these terms were three KARs (Grik2a, Grik2b, and Grik2c) found to be highly expressed in the ovipositors of older females (Fig 1E and 1F). Investigation of tissue expression revealed specific expression of Grik2b and Grik2c in the ovipositor of gravid females (Fig 1G–1I). KARs, as a class of iGluRs, are

believed to mediate the actions of glutamate [27]. However, apart from their effect on spectral preference behavior in *Drosophila* [22,23], there is limited research on the function of KARs in insects. Given their high expression in the ovipositor of gravid females, we hypothesize that they may influence the oviposition behavior of *B. dorsalis*.

## Ovipositor NMJs expressed Grik2b and Grik2c regulate oviposition behavior of gravid female

To validate the function of the receptors, we first conducted a comparative sequence analysis with that of *Drosophila melanogaster*. A maximum likelihood phylogenetic analysis utilizing amino acid sequences of KARs identified from *D. melanogaster* and other Tephritidae species revealed that three KARs in *B. dorsalis* clustered into two major families (Fig 2A), with Grik2b and Grik2c classified within the flyNMJ GluR group [28]. Amino acid alignment demonstrated that both Grik2b and Grik2c possess two polypeptides of the LBD (S1 and S2), exhibiting high similarity to those found in other insect species (S3 and S4 Figs). A sectional examination of the microstructure of the ovipositor revealed muscle cells enveloped by neurons (S5A Fig). Immunofluorescence analysis further illustrated an intertwining arrangement between muscle fibers and nerve fibers (S5B and S5C Fig). Transmission electron microscopy (TEM) results indicated close associations between muscle cells and neuronal axons, forming NMJ structures in ovipositor (S5D Fig). In situ hybridization also confirmed expression of both Grik2b and Grik2c in muscle cells at the NMJs of the ovipositor (Figs 2B, S5E, and S5F). Then, a wooden cage (S6 Fig) was utilized to assess the oviposition behavior of females. Following RNAi-induced decrease in the expression of Grik2b or Grik2c in the ovipositor 72 h after dsRNA injection (without altering other gene expression levels) (S7A–S7D Fig), there was no influence on the motion of the females (S8 Fig). However, the frequency and duration of ovipositor extrusion significantly decreased, consequently reducing the number of eggs laid (Fig 2C–2E and S1 and S2 Videos). Ovarian dissection revealed that knockdown of Grik2b/c did not lead to a reduction in the egg count within the female ovaries (S7E Fig). Furthermore, a notable decrease in oviposition preference to the fruit containing the gut bacteria (*Citrobacter* sp.) was also observed in females with knocked down Grik2b or Grik2c (Fig 2F). Additional multiple non-overlapping RNAi constructs of Grik2b or Grik2c also showed the concordant phenotypes (S7F–S7I Fig), making it highly unlikely that the effect is due to an off-target effect. However, no effect on egg-laying was observed in females with Grik2a knockdown (S7J–S7L Fig). To determine the universal influence of Grik2b and Grik2c on oviposition, we measured the oviposition attract and repel effect of 3-HA and EC, known for their oviposition attraction and repellence to females [25,26], in Grik2b and Grik2c-knocked down females. The results showed a significant decrease in the number of eggs laid (Fig 2G), and the oviposition attraction effect of 3-HA was abolished in both Grik2b and Grik2c knocked down gravid females (Fig 2H). Similarly, for EC, a significant decrease in the number of eggs laid was observed (Fig 2I), resulting in significant reduction in oviposition repellent effects in both Grik2b and Grik2c knocked down gravid females (Fig 2J). In the transcriptome, we also noted a marked high expression of genes associated with muscle development and contraction in the female ovipositor. Knockdown of these genes resulted in a significant reduction in egg-laying (S9A Fig). However, Grik2b/c knocking down did not alter the expression of these muscle development- and contraction-related genes (S9B–S9E Fig), suggesting that the effect of Grik2b/c on oviposition is independent of the transcriptional regulation of these muscle-associated genes.

## Glutamate acts as a neurotransmitter binding to Grik2b and Grik2c

As a major excitatory neurotransmitter in animals [29,30], glutamate's actions can be mediated by the KARs [19]. We hypothesized that glutamate may be the ligand of Grik2b and Grik2c. To verify this, the homology modeling of LBDs of the Grik2b and Grik2c were builded (S10A and S10C Fig). Then the binding properties of Grik2b and Grik2c to glutamate were predicted with Autodock Vina. Model construction quality evaluation results showed 99.5% and 99.2% amino acids were in the optimal and suboptimal region in the Ramachandran Plot for Grik2b and Grik2c, respectively (S10B and S10D Fig), indicating the constructed protein models could be used for docking analysis. Docking results showed that hydrogen bonds formed between glutamate and the THR-57, GLY-58, THR-87, and ARG-92 residues of Grik2b with the affinity

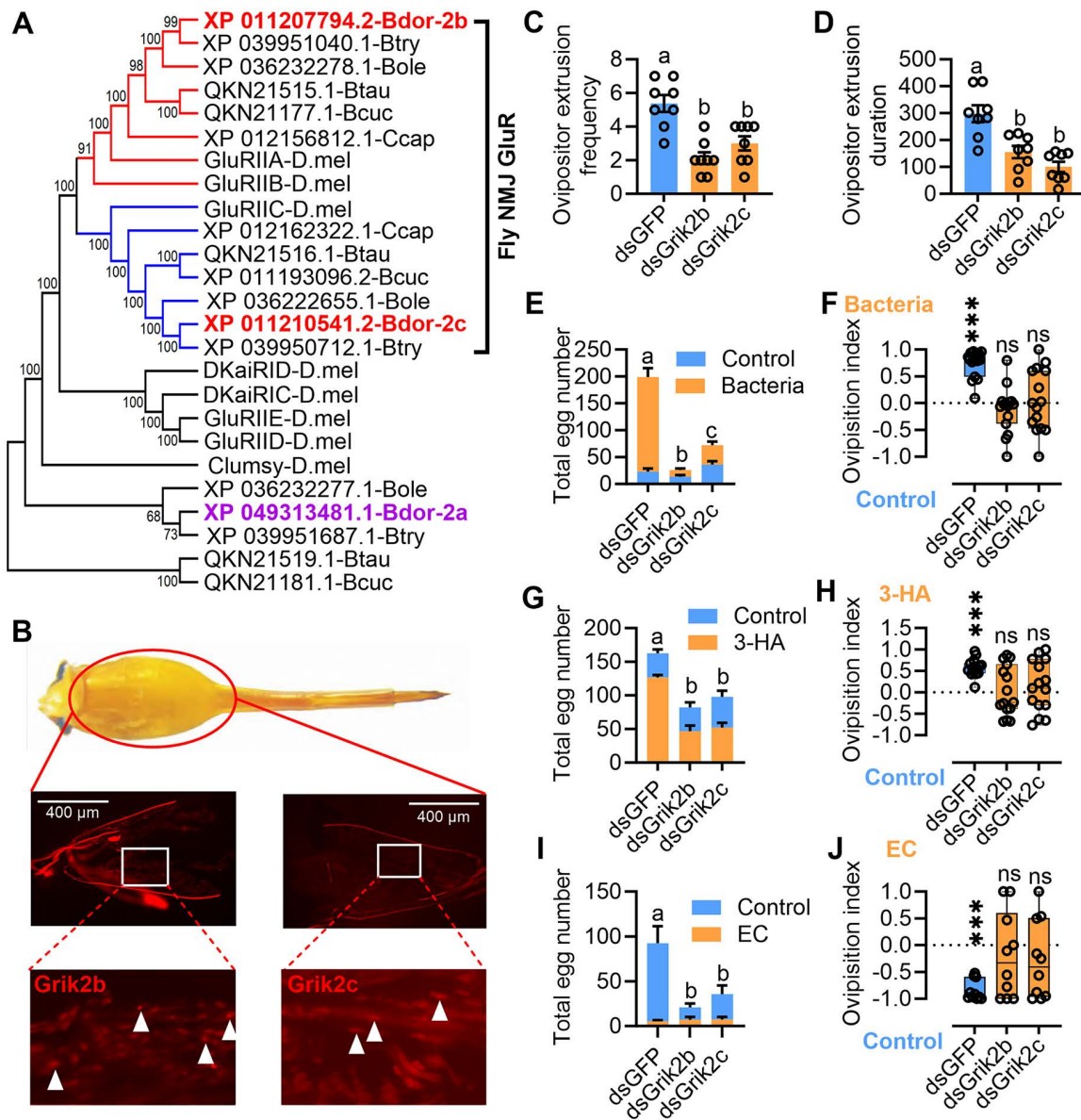

**Fig 2. Grik2b and Grik2c exert an influence on oviposition. (A)** A maximum likelihood topology tree reveals conserved clades for Grik2a, Grik2b, and Grik2c of *B. dorsalis* with *Drosophila*. **(B)** Grik2b and Grik2c identified in the muscle cells enriched oviscape basal of the ovipositor by Fluorescence in situ hybridization (FISH). The Cy3 (red) was used to label Grik2b and Grik2c. The red signal of the receptor has been demarcated with white triangles. **(C)** Influence of Grik2b/c knockdown on ovipositor extrusion frequency ($n = 8$, $F_{(2,21)} = 15.45$, $P < 0.0001$, Ordinary one-way ANOVA). **(D)** Influence of Grik2b/c knockdown on ovipositor extrusion duration ($n = 8$, $F_{(2,21)} = 16.54$, $P < 0.0001$, Ordinary one-way ANOVA). **(E)** Total eggs laid by gravid females to a gut bacteria strain added fruit ($n = 15$, $F_{(2, 84)} = 56.96$, $P < 0.001$, Two-way ANOVA). **(F)** Oviposition preference of Grik2b and Grik2c knocked down females to a gut bacteria strain added fruit (dsGFP: $n = 15$, $P < 0.001$; dsGrik2b: $n = 15$, $P = 0.297$; dsGrik2c: $n = 15$, $P = 0.74$, Paired sample student *t* test). **(G)** Total eggs laid by gravid females *t*o 3-HA added fruit (1 mg/g) ($n = 8$, $F_{(2, 42)} = 13.72$, $P < 0.0001$, Two-way ANOVA). **(H)** Oviposition preference of Grik2b and Grik2c knocked down females to 3-HA added fruit (1 mg/g) (dsGFP: $n = 15$, $P < 0.001$; dsGrik2b: $n = 15$, $P = 0.292$; dsGrik2c: $n = 15$, $P = 0.222$, Paired sample student *t* test). **(I)** Total eggs laid by gravid females to EC added fruit (1 mg/g) ($n = 10$ and 11, $F_{(2, 56)} = 9.667$, $P = 0.0002$, Two-way ANOVA). **(J)** Ovipos*i*tion preference of Grik2b and Grik2c knocked down females to EC added fruit (1 mg/g) (dsGFP: $n = 10$, $P < 0.001$; dsGrik2b: $n = 10$, $P = 0.457$; dsGrik2c: $n = 10$, $P = 0.265$, Paired sample student *t* test). dsGFP represents the control. The data underlying *t*his figure can be found in S2 Data.

being −5.2 kcal/mol (Fig 3A and S1 and S2 Tables). For Grik2c, hydrogen bonds formed between glutamate and the ARG-65, TYR-66, THR-95, ARG-98, and GLY-141 residues of Grik2c with the affinity being −4.6 kcal/mol (Fig 3B and S1 and S2 Tables). To further explore the ligand-binding ability of Grik2b and Grik2c to glutamate, we performed the proteolysis protection assays using the purified Grik2b and Grik2c LBDs expressed as soluble proteins. The results showed that, as expected, 8 mM glutamate effectively prevented digestion of LBDs by trypsin (Fig 3C and 3D). Then, microscale thermophoresis (MST) and surface plasmon resonance (SPR) were further applied to test the binding ability of Grik2b and Grik2c LBDs to glutamate. The results also showed that the Grik2b and Grik2c LBDs could bind to glutamate efficiently (Fig 3E and 3F). Moreover, *Xenopus* oocytes expressed Grik2b and Grik2c yielded stronger responses to glutamate (Fig 3G–3J), though the control *Xenopus* oocytes showed response to the glutamate. Because glutamate dehydrogenase (GluDH) and glutamate synthase (GluSN) are involved in producing glutamate in insects [31,32], we further investigated the influence of glutamate on oviposition behavior by knocking down expression of GluDH and GluSN, respectively. The results showed that expression of GluDH and GluSN in ovipositor significantly decreased 72 h after dsRNA injection (S11 Fig). Accordingly, glutamate levels in ovipositor decreased significantly (Fig 3K). Behavior assays showed that knocking down expression of GluDH and GluSN had no influence on motion of the females (S8 Fig), while number of eggs laid and oviposition preference to the gut bacteria added fruit decreased significantly in GluDH and GluSN-knocked down females (Fig 3L and 3M). These results demonstrated that glutamate is the neurotransmitter activating Grik2b and Grik2c and influences oviposition of the females.

**Expression of Grik2b and Grik2c in GNs controls the oviposition preference**

The expression of the receptor in the NMJs of the ovipositor could plausibly affect the egg-laying quantity by regulating muscle contraction. Nevertheless, the receptor also governs egg-laying preference, suggesting that the receptor might be expressed in the upstream neurons that modulate egg-laying preference. The response of Grik receptors to glutamate, along with their documented co-expression with markers for GNs in comparable systems [33,34], led us to investigate their potential presence and role in our model. Although qPCR results showed low expression levels of Grik2b/2c in thoracic tissues (Fig 1H and 1I), we speculate that Grik2b/2c may be expressed only in a specific subset of GNs within the thoracic region to regulate oviposition preference. Thus, FISH was utilized to determine the expression localization of Grik2b, Grik2c, and vesicular glutamate transporter (Vglut) in the brain and ventral nerve cord of females. The results indicated that Grik2b, Grik2c, and Vglut were co-expressed in the GNs of the brain (Fig 4A and 4B) and ventral nerve cord (Fig 4C and 4D) of females, though not detected by qPCR in the head and thorax (Fig 1H and 1I). However, Vglut did not colocalize with Grik2b and Grik2c in the oviscape basal of ovipositor (Fig 4E and 4F). Tissue expression investigation also revealed that Vglut was mainly expressed in the CNS containing parts of the females (Fig 4G). Given that Vglut is a specific marker gene for GNs [35], it can be concluded that Grik2b and Grik2c are also expressed in GNs and might play a role in influencing oviposition preference. RNAi assays demonstrated that knockdown of Vglut expression could eliminate the oviposition preference for gut bacteria-added fruit (Fig 4H and 4I), while no effect on the total number of eggs laid was noted (Fig 4J), suggesting that GNs regulate the oviposition preference of females. The results of calcium ion signal detection in the ventral nerve cord also indicated that silencing the expression of Grik2b, Grik2c, and Vglut could lead to attenuation of calcium signals in GNs (Fig 4K and 4L). Furthermore, 12 h after the injection of dsRNA targeting Grik2b or Grik2c into the thorax, it could specifically cause a reduction in the expression of Grik2b or Grik2c in the CNS, while have no impact on the expression in the ovipositor (S12A and S12B Fig). Correspondingly, the female's oviposition preference for gut bacteria added fruit would decrease (Fig 4M), while the total number of eggs laid would remain unchanged (Fig 4N). However, 24 h after the injection of dsRNA targeting Grik2b or Grik2c into the thorax, reduction in the expression of Grik2b or Grik2c was induced in both CNS and ovipositor (S12C and S12D Fig). Correspondingly, the female's oviposition preference and total number of eggs for gut bacteria added fruit changed significantly (S12E and S12F Fig). Furthermore, 6 h after the injection of Grik2b or Grik2c dsRNA into the abdomen only results in the downregulation of Grik2b or Grik2c

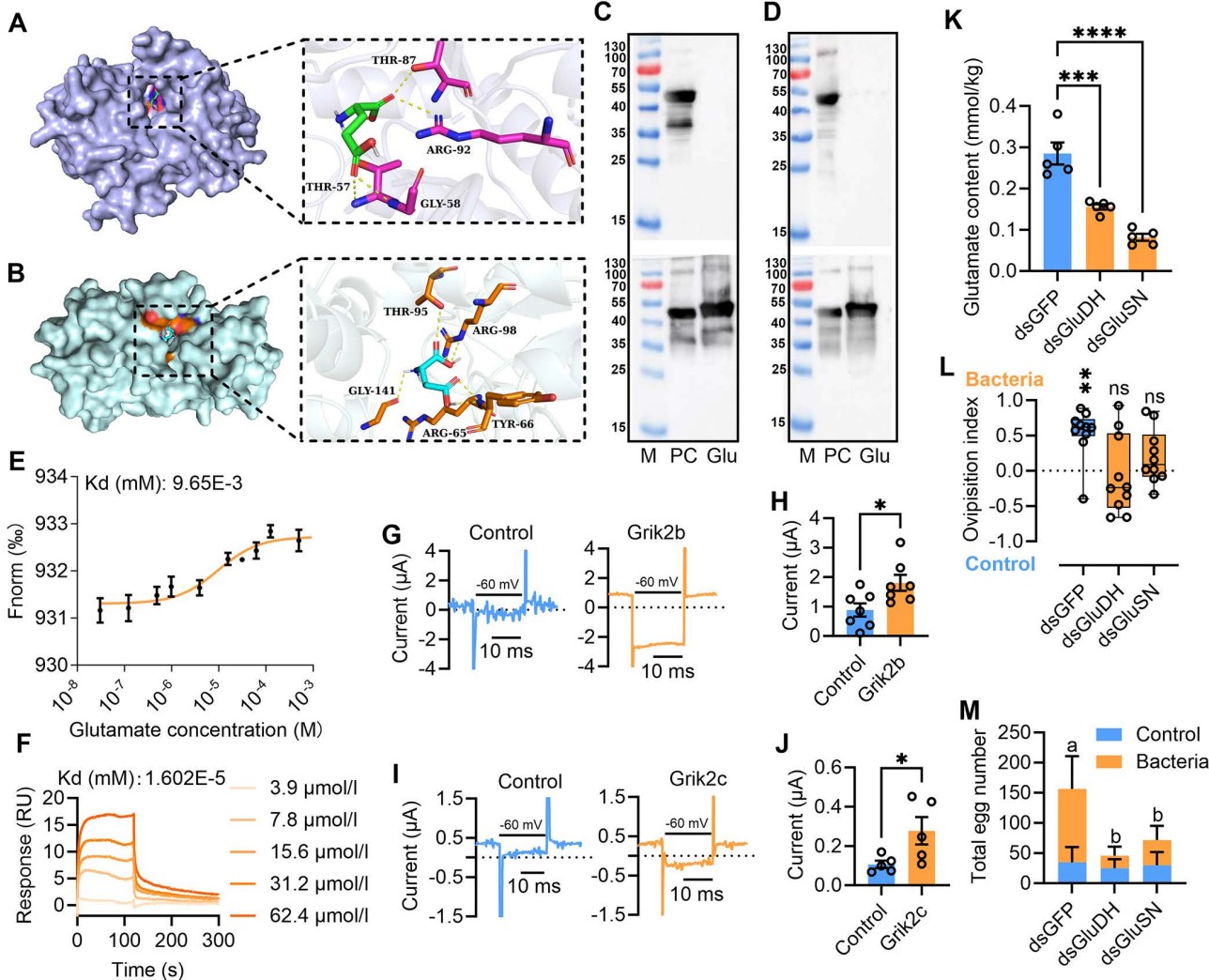

**Fig 3. Glutamate acts as a neurotransmitter binding to Grik2b and Grik2c and influencing oviposition. (A)** Binding sites prediction of Grik2b to glutamate by Autodock Vina. **(B)** Binding sites prediction of Grik2c to glutamate by Autodock Vina. **(C)** Proteolysis protection assays for digestion of Grik2b LBD for 20 min by trypsin at a 1:20 ratio, with the glutamate added at a concentration of 1 mM (up) and 8 mM (down), respectively. **(D)** Proteolysis protection assays for digestion of Grik2c LBD for 20 min by trypsin at a 1:20 ratio, with the glutamate added at a concentration of 1 mM (up) and 8 mM (down), respectively. **(E)** MST showing binding affinity of glutamate to Grik2b LBD. Graphs show the MST data fitted with Kd model binding curves for Grik2b LBD and glutamate it bound. Each curve and data point represents an average of three independent experiments. **(F)** SPR showing binding responses of glutamate to Grik2c LBD. Glutamate was injected over the chip surface containing Grik2c LBD. The purified Grik2c LBD (20 μg/ml) was coated on a CM-5 chip and analyzed for interactions by determining its affinity coefficients with glutamate. **(G)** Example traces of responses to 3 mM glutamate at −60 mV for oocytes injected with cRNA for Grik2b, all treated with concanavalin A and recorded on the same day from the same batch of oocytes. The peaks on both sides of the recorded current are the capacitive currents produced during the charging and discharging of oocytes. **(H)** Currents comparison between responses to 3 mM glutamate at −60 mV for oocytes injected with cRNA for Grik2b and the control ($n=7$, $P=0.0222$, Independent sample student $t$ test). **(I)** Example traces of responses to 3 mM glutamate at −60 mV for oocytes injected with cRNA for Grik2c, all treated with concanavalin A and recorded on the same day from the same batch of oocytes. The peaks on both sides of the recorded current are the capacitive currents produced during the charging and discharging of oocytes. **(J)** Currents comparison between responses to 3 mM glutamate at −60 mV for oocytes injected with cRNA for Grik2c and the control ($n=5$, $P=0.0453$, Independent sample student $t$ test). **(K)** Ovipositor glutamate content comparison between GluDH (GluSN) knocked down females and the control ($n=5$, $F_{(2,12)}=38.62$, $P<0.0001$, Ordinary one-way ANOVA). **(L)** Oviposition preference of GluDH(GluSN) knocked down females to the gut bacteria strain added fruit (dsGFP: $n=10$, $P=0.001$; dsGluDH: $n=10$, $P=0.72$; dsGluSN: $n=10$, $P=0.13$, Paired sample student $t$ test). **(M)** Total eggs laid by GluDH (GluSN) knocked down females ($n=10$, $F_{(2,54)}=14.11$, $P<0.0001$, Two-way ANOVA). In the figures, dsGFP represents the control. In (C) and (D), M: Marker (the numbers on left were the weight numbers (KDa)); PC: Positive control; Glu: Glutamate added. In **(G–J)**, Control means the oocytes not injected with cRNA of Grik2b and Grik2c. The current value indicates the disparity in current recorded under 0 and −60 mV potential. The non-specific endogenous current generated in the control might be attributed to the ion channels possessed by oocytes. The data underlying this figure can be found in S3 Data.

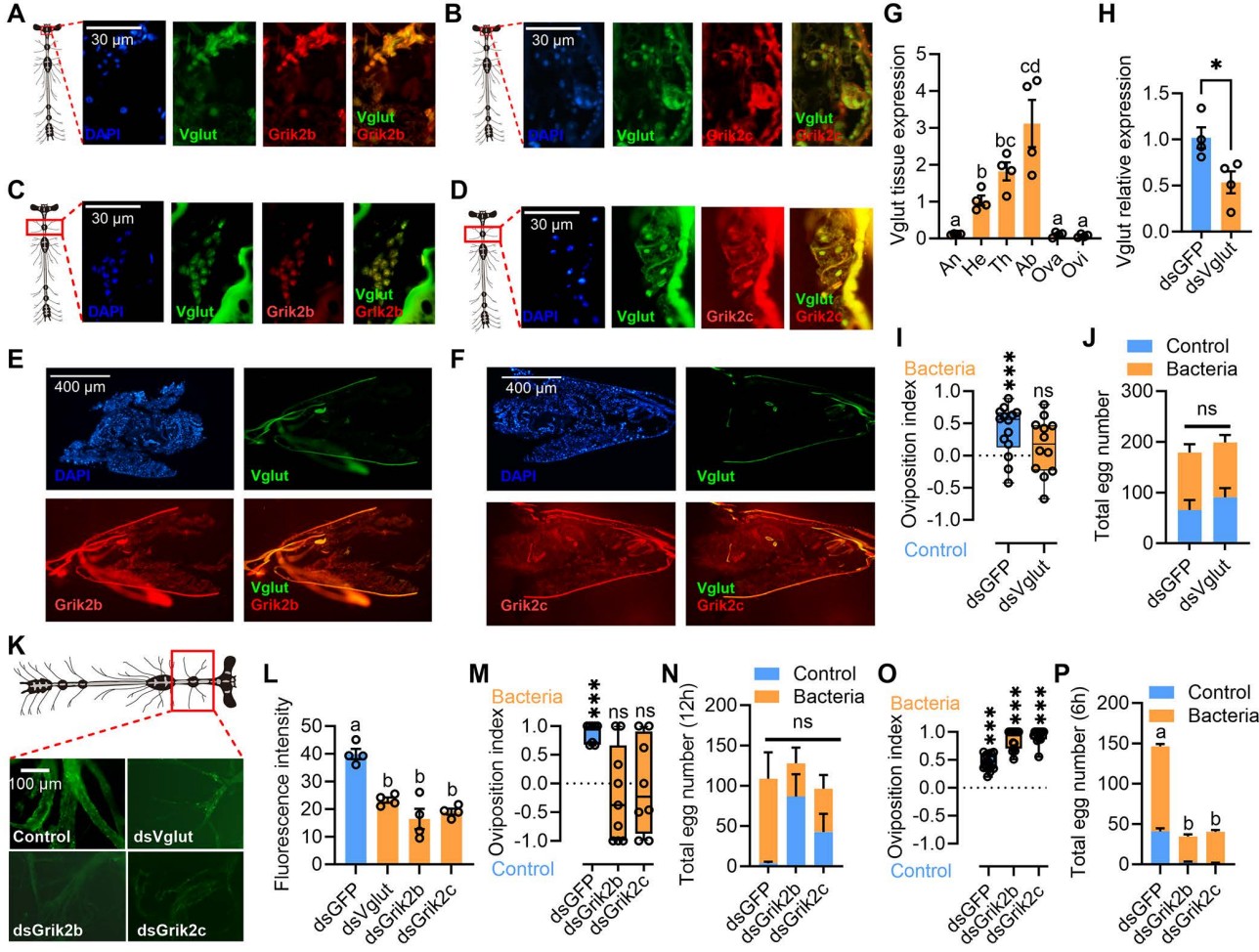

**Fig 4. GNs expressing Grik2b and Grik2c control oviposition preference to the gut bacteria. (A, B)** Colocalization between Vglut/Grik2b/Grik2c mRNA signals and nuclear signal (DAPI) in female brain. **(C, D)** Colocalization between Vglut/Grik2b/Grik2c mRNA signals and nuclear signal (DAPI) in female cephalothoracic ganglion. **(E, F)** Vglut mRNA signal did not colocalize with either the Grik2b/Grik2c mRNA signals or the nuclear signals (DAPI) in oviscape basal of ovipositor. **(G)** Tissue expression of Vglut in gravid females ($n=4$, $F_{(5,18)}$ = 18.51, $P<0.0001$, Ordinary one-way ANOVA). **(H)** RNAi efficiency of Vglut in CNS 72 h after dsRNA injection ($n=4$, $P=0.0262$, Independent sample student $t$ test). **(I)** Oviposition preference of Vglut knocked down females to the gut bacteria strain added fruit (dsGFP: $n=14$, $P<0.0001$; dsVglut: $n=12$, $P=0.27$; Paired sample student $t$ test). **(J)** Total eggs laid by Vglut knocked down females ($n=14$ and 12, $F_{(1,48)}$ = 0.7331, $P=0.3961$, Two-way ANOVA). **(K)** Cephalothoracic ganglion calcium ion signal of females with Vglut, Grik2b, and Grik2c knocked down. **(L)** Cephalothoracic ganglion calcium ion signal comparison between females with Vglut, Grik2b, and Grik2c knocked down ($n=4$, $F_{(3,12)}$ = 23.04, $P<0.0001$, Ordinary one-way ANOVA). **(M)** Oviposition preference to the gut bacteria strain added fruit of females with dsRNA of Grik2b and Grik2c injected into thorax for 12 h (dsGFP: $n=8$, $P<0.001$; dsGrik2b: $n=9$, $P=0.48$; dsGrik2c: $n=8$, $P=0.89$, Paired sample student $t$ test). **(N)** Total eggs laid by females with dsRNA of Grik2b and Grik2c injected into thorax for 12 h to a gut bacteria strain added fruit ($n=8,9$ and 8, $F_{(2,44)}$ = 5.411, $P=0.0079$, Two-way ANOVA). **(O)** Oviposition preference to the gut bacteria strain added fruit of females with dsRNA of Grik2b and Grik2c injected into abdomen for 6 h (dsGFP: $n=15$, $P<0.001$; dsGrik2b: $n=15$, $P<0.001$; dsGrik2c: $n=15$, $P<0.001$, Paired sample student $t$ test). **(P)** Total eggs laid by females with dsRNA of Grik2b and Grik2c injected into abdomen for 12 h to a gut bacteria strain added fruit ($n=15$, $F_{(2,56)}$ = 29.37, $P<0.001$, Two-way ANOVA). In (A–F), the DAPI and FAM (green) were used to label the nucleus and Vglut; Cy3 (red) was used to label Grik2b and Grik2c. Fluo-4 AM (green) was used to detect the intracellular calcium ion in (K). In (N) and (P), reduction in overall egg-laying compared with other experiments may be caused by the physiological stress from the injection procedure combined with the shorter recovery period. The data underlying this figure can be found in S4 Data.

in the ovipositor (S12G and S12H Fig). Consequently, the total number of eggs was significantly affected, other than the oviposition preference (Fig 4O and 4P). However, 12 h after the injection of dsRNA targeting Grik2b or Grik2c into the abdomen, reduction in the expression of Grik2b or Grik2c was induced in both CNS and ovipositor (S12I and S12J Fig). Correspondingly, the female's oviposition preference and total number of eggs for gut bacteria added fruit changed significantly (S12K and S12L Fig). These findings suggest that the GNs and ovipositor expressing Grik2b and Grik2c are accountable for the oviposition preference and egg number laid to gut bacteria-infected fruit, respectively.

### Inhibitors modulates oviposition by interacting with the KARs

We further screened the KAR inhibitors and assessed their influence on oviposition. In studies on vertebrates, D-2-Amino-5-phosphonovaleric acid (D-AP5) [28] and 2, 3-dihydroxy-6-nitro-7-sulfamoyl-benzo (F) quinoxaline (NBQX) [36] are selective antagonists for NMDA receptors and AMPA/KARs, respectively [36,37]. Research in *Drosophila*, however, has shown that KARs can be inhibited by D-AP5 [28]. Phylogenetic analysis reveals that Grik2b and Grik2c share high similarity with KARs in *Drosophila* (Fig 2A). Therefore, we selected these two antagonists to assess their inhibitory effects on Grik2b/c. The docking results revealed the hydrogen bonds existed between the selected ligands and Grik2b (Fig 5A and 5B and S2 Table). Among the identified binding sites, ARG-92, THR-87, and/or THR-57 were found to be common binding sites of glutamate to Grik2b (S2 Table). Additionally, hydrogen bonds were also observed between the selected ligands and Grik2c (Fig 5C and 5D and S2 Table). TYR-66, THR-95, ARG-98, and/or GLY-141 were the common binding sites as glutamate to Grik2c (S2 Table). D-AP5 and NBQX exhibited higher affinities for Grik2b and Grik2c than glutamate (S1 Table). Proteolysis protection assays revealed that at concentrations of 4 and 8 mM, D-AP5 and NBQX prevented trypsin digestion of Grik2b LBDs (Fig 5E). For Grik2c, only NBQX showed prevention of trypsin digestion at a concentration of 4 and 8 mM (Fig 5F). MST and SPR revealed that Grik2b LBDs exhibited binding capability with all selected inhibitors (Fig 5G and 5H), albeit at lower affinities compared to glutamate (Fig 3E). Similarly, for Grik2c, all inhibitors demonstrated binding to the LBDs (Fig 5I and 5J) with affinities lower than glutamate (Fig 3F). Upon injection into the abdomen of gravid females, all inhibitors caused significant decrease in oviposition preference and number of eggs laid towards the gut bacteria added fruit (Fig 5K and 5L). These results suggest that the selected inhibitors can bind to Grik2b and Grik2c, thereby influencing oviposition.

### Discussion

The insect olfactory system, with its amenability to genetic manipulation, has facilitated increasingly profound exploration of the molecular and cellular basis of odorant-driven behaviors, including the oviposition preference [10,26,38]. Our research has provided strong supporting evidence that the neural conduction mechanism underlying the oviposition by uncovering the dual role of KARs in GNs and NMJs. We have provided evidence that the GNs and ovipositor NMJs modulate oviposition (preference and number of eggs laid) by receiving excitatory glutamatergic signals via KARs (Fig 6). We now explicitly compare our findings with the studies about oviposition in *Drosophila* [17,18]. Our research in *B. dorsalis* confirms the broad importance of glutamate signaling, particularly through Grik receptors. This finding strongly corroborates discoveries in *Drosophila*, suggesting this signaling pathway may be deeply evolutionarily conserved in insect egg-laying behavior.

As a subtype of iGluRs [27], insect KARs have undergone functional validation experiments that revealed ligand binding properties distinct from those observed in vertebrates [28], which indicates the novel function of KARs in insects. Research in *D. melanogaster* and *Schistocerca gregaria* has demonstrated that iGluRs mediate synaptic transmission at the NMJ, leading to postsynaptic membrane depolarization [21,20]. Subsequently, analogous glutamate receptors were identified in the median neurons of *Periplaneta americana* and *Locusta migratoria* [39,40]. Despite the recent discovery of many iGluRs in insects, their functional properties remain inadequately validated, particularly concerning insect behavior. Study has indicated that the KAR is significantly expanded in insects and the function of these receptors is only

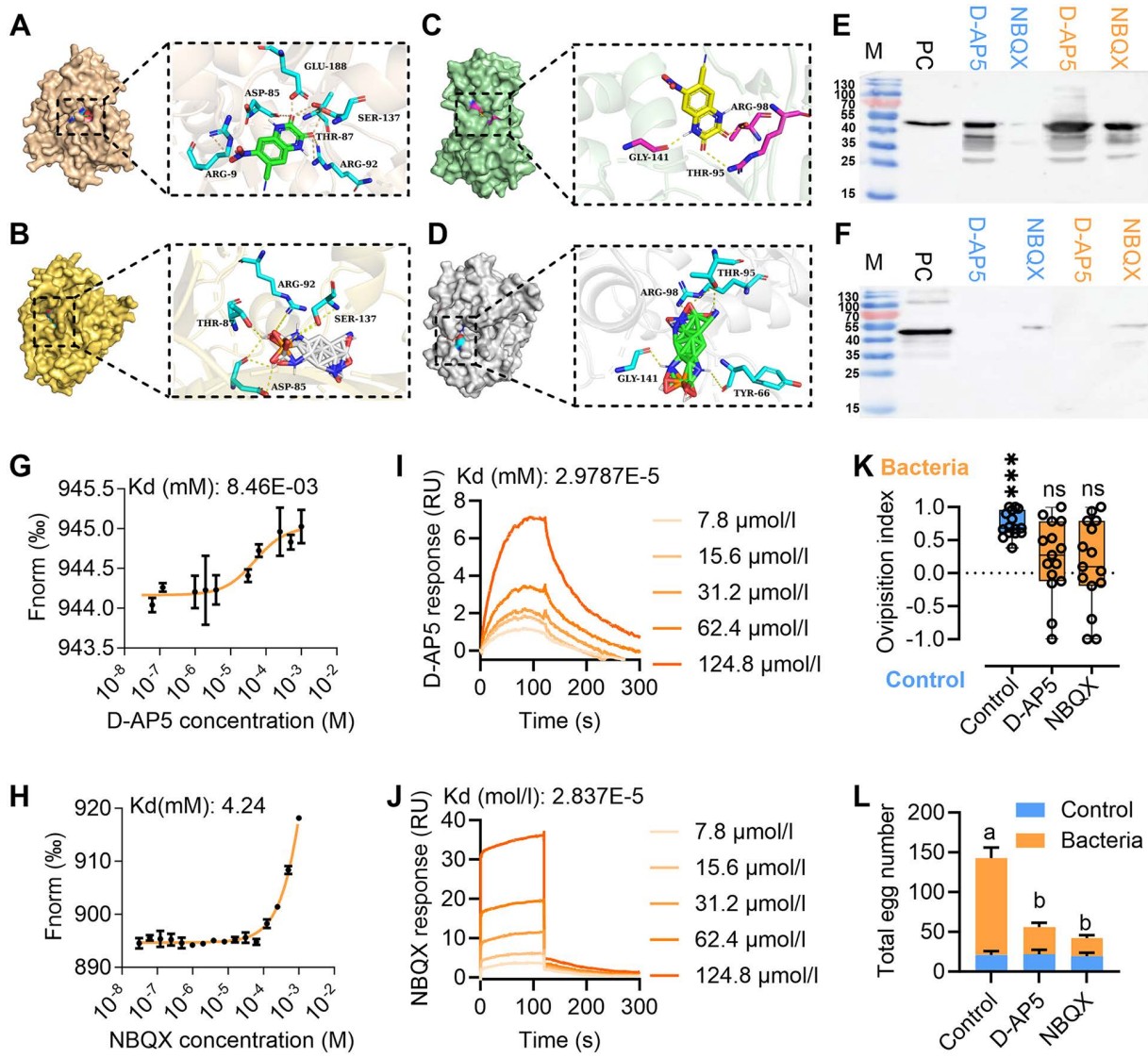

**Fig 5. KAR inhibitors modulate oviposition through competitive binding to Grik2b and Grik2c. (A, B)** Molecular docking result between Grik2b and D-AP5 or NBQX, respectively. **(C, D)** Molecular docking result between Grik2c and D-AP5 or NBQX, respectively. **(E)** Proteolysis protection assays for digestion of Grik2b LBDs for 20 min by trypsin at a 1:20 ratio, with the KAR inhibitors added at a concentration of 4 mM (blue) and 8 mM (yellow), respectively. **(F)** Proteolysis protection assays for digestion of Grik2c LBDs for 20 min by trypsin at a 1:20 ratio, with the KAR inhibitors added at a concentration of 4 mM (blue) and 8 mM (yellow), respectively. **(G, H)** MST showing binding affinity of D-AP5 and NBQX to Grik2b LBD, respectively. **(I, J)** SPR showing binding responses ofD-AP5 and NBQX to Grik2c LBD, respectively. **(K)** Oviposition preference of the inhibitors injected females to the gut bacteria strain added fruit (Control: $n = 15$, $P < 0.001$; D-AP5: $n = 15$, $P = 0.15$; NBQX: $n = 15$, $P = 0.46$, Paired sample student $t$ test). **(L)** Total eggs laid by the KAR inhibitors injected females ($n = 15$, $F_{(2,84)} = 30.32$, $P < 0.001$, Two-way ANOVA). In (E) and (F), M: Marker (the numbers on left were the weight numbers (KDa)), PC: Positive control. For (G and H), graphs show the MST data fitted with Kd model binding curves for Grik2b LBD and the ligands it bound. Each curve and data point represents an average of three independent experiments. For (I and J), the ligands were injected over the chip surface containing Grik2c LBD. The purified Grik2c LBD (20 μg/ml) was coated on a CM-5 chip and analyzed for interactions by determining its affinity coefficients with the ligands. The data underlying this figure can be found in S5 Data.

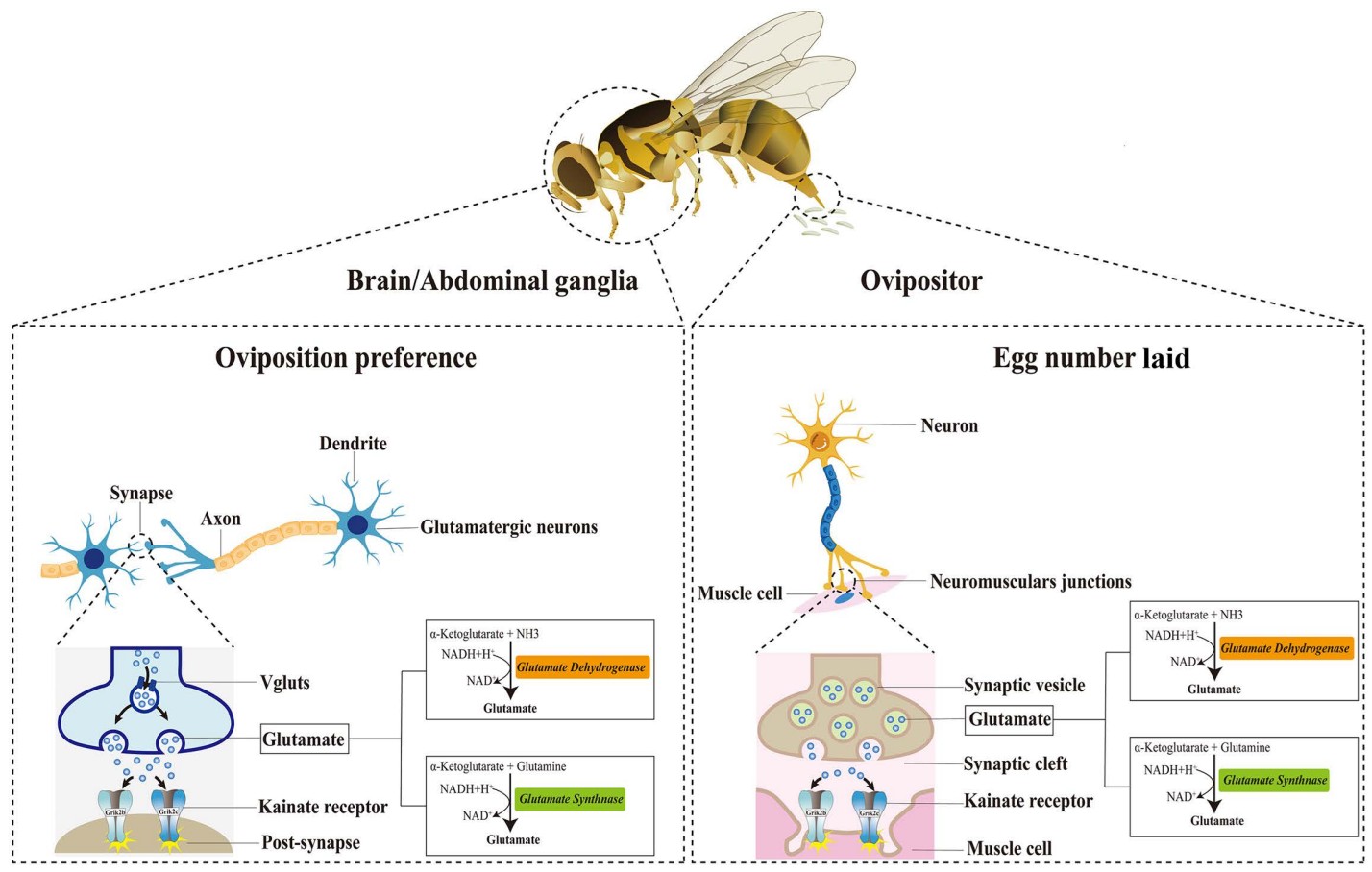

**Fig 6. Mechanism diagram of the regulation of oviposition behavior by KARs.**

hypothesized to be involved in the CNS and NMJ of insects [28]. To date, research on the biological function of the KAR in insects is even scarcer, with only two systematic reports documenting the expression of the KAR in the eyes of *Drosophila* and its involvement in light perception [22,23]. Our study further disclosed that KARs in the GN and ovipositor NMJ regulate oviposition, further confirming the new function and significant role of the KAR.

The female reproductive systems of insects are characterized by extensive muscularization and intricate innervation by neuronal processes. Research on *Locusta migratoria* and *D. melanogaster* has demonstrated that egg-laying behavior is regulated by the CNS's control over relevant musculature [41,42]. Neurons within the CNS can modulate reproductive tissues associated with oviposition by releasing neurotransmitters at the NMJ. For example, octopamine influences the expulsion of eggs from the ovaries by regulating contractions and expansions of the reproductive tract [11,43,44]. Similarly, glutamate has been shown to localize with numerous motor neurons responsible for driving muscle contraction in insects [45]. Given that KARs are classified within the NMJ GluR group [28], their disruption in muscle contractions at the NMJs of the ovipositor may elucidate their observed influence on number of eggs laid.

Oviposition preference for volatiles in insects has been extensively studied [5,10,46] by emphasizing the critical role of the olfactory system [38,47–49]. However, the understanding of the neural conduction mechanism of the oviposition remains rather restricted. In our study, we demonstrate that KARs in GNs transmit the oviposition preference signal. In *D. melanogaster*, it has been revealed that the KAR mediates the preference for ultraviolet light [22]. Our research further

demonstrated that the KARs in the CNS and GNs play role in regulating oviposition preference, which suggests that the KARs might have a functional role in neurons (such as GNs) responsible for transmitting signals associated with oviposition preference to bacteria and volatiles.

Nevertheless, this study also raises additional questions that warrant further investigation. Given the impact of glutamate on oviposition behavior and the observed influence of RNAi on GluSN genes in modulating this behavior, it is essential to explore the role of the glutamate synthesis pathway in ovipositor function during oviposition. This study reveals the fundamental regulatory role of glutamatergic signaling in the overall excitability of the VNC through population calcium imaging. However, due to the lack of cell-type-specific genetic tools in non-model organisms, we are currently unable to resolve finer temporal dynamics in this regulatory process (e.g., the precise onset, oscillations, or propagation patterns of neural activity). In the future, with the development of cross-species genetic tools and high-speed imaging technologies, elucidating these millisecond-level dynamic characteristics will be crucial for understanding how glutamate precisely shapes the temporal structure of behavioral outputs. The gold standard for establishing a functional separation between oviposition preference and egg quantity is to conduct the rescue experiment. For instance, this can be achieved by ubiquitously knocking out or knocking down a gene and then restoring its expression solely in the CNS and GNs. Demonstrating that the preference is restored while the egg quantity remains unaffected would provide evidence. Nevertheless, at present, such tools are not yet available for *B. dorsalis*. While our study demonstrates that pharmacological inhibition of Grik receptors via thoracic injection effectively suppresses egg-laying, this method is impractical for field applications. Future work is essential to develop deployable delivery strategies, such as formulating stable and orally active Grik antagonists that can be incorporated into bait stations or sprays. The identification of Grik as a valid target in this study provides a critical foundation for these applied endeavors.

## Materials and methods

### Insect rearing

The *B. dorsalis* strain was obtained from a carambola orchard in Guangdong Province, China, in April 2022. The flies were reared under laboratory conditions at $27 \pm 1$ °C, $70\% \pm 5\%$ relative humidity, and a light:dark photoperiod of 12:12 h. Larvae were provided with a maize-based diet consisting of 150 g corn flour, 150 g banana, 0.6 g sodium benzoate, 30 g yeast, 30 g sucrose, and 30 g paper towel mixed with hydrochloric acid and water. Newly emerged adult flies were collected and placed in a 60 cm * 60 cm * 60 cm insect-rearing cage with an artificial diet (yeast:sugar = 1:3) and access to water.

### Oviposition assays

In our previous study [25], we have provided a detailed description of the methodological approach employed for oviposition experiments. Briefly, two 50 ml centrifuge tubes were positioned within a wooden cage (dimensions: length:width:height = 60 cm:60 cm:60 cm) for conducting 2-choice experiments. Both the interior of the cage and the centrifuge tubes underwent thorough disinfection. Fresh guava was sterilized using 75% ethanol and subsequently ground into puree with a sterile grinder. The resulting puree (2 g) was then mixed with 100 µL sterile water, as well as with another portion of puree (2 g) mixed with 100 µL gut bacteria ($0.8 * 10^8$ cfu/mL) in sterile water, before being placed in a cage containing 10 gravid female flies. After an interval of 2 h, the number of eggs present in the tubes was tallied, and an oviposition index was calculated using the equation $(O - C)/(O + C)$, where O represents the quantity of eggs in the treatment group and C represents that in the control group. The identical approach was employed to assess the oviposition preference and avoidance behavior of the flies towards 3-HA and EC added fruit (1 mg/g), respectively.

### Transcriptome sequencing and analysis

Female ovipositors were collected at 0, 3, 6, 9, and 12 days post-eclosion. Total RNA was extracted from 20 ovipositors using Trizol Reagent (Thermo Fisher Scientific, United States) following the manufacturer's protocol. Additionally, RNA

samples at each time point were replicated five times. The paired-end RNA-seq library was constructed according to Illumina's library construction protocol and sequenced on the Illumina HiSeq2000 platform following standard protocols. Raw sequencing reads from each cDNA library were processed to eliminate adaptors, low-quality sequences ($Q < 20$), and contaminated reads. Paired-end resulting reads were aligned to the *B. dorsalis* reference genome using HISAT2. 2.4 with "-rna-strandness RF" and other parameters set as a default [50] to obtain unique mapped reads. Transcript expression abundance was calculated and normalized into FPKM (Fragments Per Kilobase per Million fragments) using String-Tie v1.3.1 software [51]. Differentially expressed genes (DEGs) were identified based on a fold change ≥ 2.0 and a false discovery rate (FDR < 0.05) utilizing DESeq2 software [52]. The ovipositor transcriptome data was analyzed using gene expression pattern analysis to cluster genes with similar expression patterns. The expression pattern of the DEGs was obtained by clustering the normalized data of each sample with Short Time-series Expression Miner software ($p < 0.05$ were considered significant profiles) [53].

## Validation of gene expression by quantitative real-time PCR (qRT-PCR)

To compare the expression of selected genes at different developmental times and in different tissues, qRT-PCR was performed. Ovipositors were collected at 0, 3, 6, 9, and 12 days after eclosion. Antennae, heads (without antennae), thoraxes, abdomens, ovaries, and ovipositors of 12-day-old females were collected separately. Total RNA extraction was then carried out from the collected samples. The corresponding cDNAs were synthesized from total RNA samples using the One-Step gDNA Removal and cDNA Synthesis SuperMix Kit (TransGen Biotech, Beijing, China) according to the manufacturer's instructions. Expression of genes at different time points and tissues was verified by qRT-PCR using a reaction mixture containing 0.8 µL cDNA, 0.4 µL of each gene-specific primer (400 nM), 5 µL of 2 × TB Green Premix Ex Taq II (Tli RNaseH Plus), and 3.4 µL Nuclease-free water in a final volume of 10 µL. The qRT-PCR for each cDNA was performed in triplicate in CFX96 Real-Time PCR Detection System with the following profile: initial denaturation at 95 °C for 30 s followed by 40 cycles of denaturation at 95 °C for 5 s, annealing/extension at 60 °C for 30 s, and finally a melting curve stage. Expression levels of the reference genes were used as endogenous references for normalization [54].

## Hematoxylin and eosin (HE) staining

The ovipositors (oviscape basal) from 12-day-old females were dissected and subsequently embedded in paraffin wax. A 4 µm section was obtained using a freezing microtome (Leica, EM, UC7/FC7, Germany). The paraffin-embedded sections were dewaxed with xylene and ethanol. Subsequently, the samples were rehydrated with xylene and ethanol, followed by staining with hematoxylin for 5 min. The sections were then rinsed with distilled water for 30 min and stained with eosin for 3 min. After washing with distilled water, dehydration and hyalinization procedures were carried out. Finally, the section was sealed using neutral resin before being analyzed and recorded under an optical microscope (Nikon Eclipse E100, Japan).

## Neuronal and muscle fibers stained by immunocytochemistry

The muscle fibers of ovipositor (oviscape basal) were dissected from 12-day-old females in PBS and fixed in 4% paraformaldehyde for 30 min at room temperature. The samples underwent three washes with PBST and were then treated with 0.1% Triton X-100 for 10 min. Following another wash in PBST, the samples were blocked in 1% BSA for 1 h at room temperature before being incubated overnight at 4°C with the primary antibody anti-HRP (1:100) (Jackson Immuno Research, United States). Subsequently, TRITC Phalloidin (diluted in PBS) (Yeasen, Shanghai, China) was applied to target the cytoskeleton for 30 min, followed by a final wash. Finally, the treated ovipositor muscles were observed using fluorescence microscopy to detect the fluorescence of neuronal and muscle fibers in the red and green channels, respectively.

## Neurons and muscle cells stained by immunocytochemistry

The ovipositors (oviscape basal) were dissected from 12-day-old females in PBS and fixed in 4% paraformaldehyde for over 24 h. The samples were then transferred to a 15% sucrose solution for dehydration for 2 days, followed by transfer to a 30% sucrose solution for an additional 2 days. The dehydrated tissue was embedded in OCT medium and longitudinally sectioned into 4 μm sections using a freezing microtome (Thermo Fisher Scientific, Cryotome E, United States). The ultrathin sections were blocked with 1% BSA for 1 h at room temperature before incubation with the primary antibody anti-HRP (1:100) (Jackson Immuno Research, United States) overnight at 4 °C. Subsequently, TRITC Phalloidin (diluted in PBS) (Yeasen, Shanghai, China) was used to label the cytoskeleton for 30 min, followed by washing of the samples. DAPI (4′,6-diamidino-2-phenylindole) was added to stain the nuclei for 5 min. Finally, the sections were observed under fluorescence microscopy in red, green and blue channels to detect neuronal fluorescence cells and muscle cells as well as nuclear staining.

## Observation of ovipositor NMJs by transmission electron microscopy (TEM)

The ovipositors (oviscape basal) dissected from 12-day-old females were fixed in 2.5% glutaraldehyde at 4 °C overnight and subsequently post-fixed in 1% osmium tetroxide for 2 h. The fixed samples were then immersed in uranyl acetate dihydrate at 4 °C overnight for staining. Following this, the samples underwent dehydration through a series of ethanol concentrations before being embedded in neutral resin. The embedded tissue was trimmed to an appropriate size using a trimmer and cut longitudinally into slices measuring 70–100 nm with an ultramicrotome. The resulting ultrathin sections were examined using a Talos L120C Transmission Electron Microscope (FEI, United States).

## Gene expression analysis via Fluorescence In Situ Hybridization (FISH)

Brain, ovipositors (oviscape basal) and ventral nerve cord were dissected from 12-day-old females and subsequently embedded in paraffin wax. A 5 μm section was cut and mounted on a slide. The sample was dewaxed with xylene for three cycles of 8 min each, followed by dehydration treatment using ethanol (100% ethanol for 5 min, then 95% ethanol for another 5 min, and finally 75% ethanol for an additional 5 min) at room temperature. The slide was then placed in phosphate-buffered saline (PBS). Subsequently, the slides were immersed in a tissue cell drilling solution (99.5 ml of 0.1 M citric acid buffer + 0.5 ml Triton X-100) at room temperature for 10 min before being washed three times with PBS. Following this step, the tissue was incubated in pepsin working solution (1 ml containing a mixture of drops with 3% citric acid plus two drops of original pepsin solution) for 20 minutes at a temperature of 37°C. Next, the tissue was covered with prehybridization working solution (buffer used for in situ hybridization maintaining salmon sperm DNA at a final concentration of 100 μg/ml) within a wet box set to 42°C for 2 h. The slide underwent washing three times using diluted SSC (0.2 × SSC; prepared by dissolving 175.3 g NaCl and 88.2 g trisodium citrate in sterilized water to adjust pH to 7.0 with 1M HCl and bringing the volume up to 1,000 ml) at room temperature for five minutes. Twenty microliters of hybrid working solution (NaCl 30 g; $Na_2HPO_4 \cdot 12H_2O$ 6 g; $NaH_2PO_4 \cdot 2H_2O$ 4 g adjusted to pH 7.2–7.6 to constant volume of 1 L), containing probe (1 μM) (S3 Table), was added onto the slide. The slide was then covered with cover glass for in situ hybridization within a wet box at 42 °C for 12–18 h. Finally, the cover glass was removed and washed three times using (2 × SSC). Lastly, 50 μL of DAPI solution (prepared by diluting the original DAPI solution at a ratio of 1:500) was applied to the slide and incubated in darkness for five minutes. Subsequently, the slides were sealed with an anti-fluorescence quenching agent and examined under a fluorescence microscope. For control, we also used sense probes to hybridize the tissues and treated the sections with RNase A before hybridization.

## Detection of calcium ion signals in ventral ganglia

Cephalothoracic ganglion was dissected from 12-day-old females, which were maintained active in pre-cooled PBS buffer. Subsequently, real-time monitoring of the calcium signal was conducted using Fluo-4 AM. Before utilization, the probe was

pre-diluted in PBS to generate a stock solution. Then, a working solution (100 µM) was prepared in PBS. The dissected ganglia were fully immersed in the Fluo-4 AM working solution and incubated for 2 h at 42 °C in a wet box. The slides were washed three times with PBS at room temperature. The slides were sealed with an anti-fluorescence quencher, and the fluorescence of Fluo-4 AM was detected under a fluorescence microscope for the determination of calcium ions. Quantitative analysis of fluorescence intensity was performed using ImageJ 1.54d (National Institutes of Health, USA). The Li algorithm in the color threshold tool was applied to select fluorescence signal areas, and the Mean Gray Value of the selected regions was analyzed.

## Synthesis and delivery of double-stranded RNA (dsRNA) via microinjection

Double-stranded RNA (dsRNA) primers targeting the genes were designed using the NCBI primer design tool, incorporating T7 promoter sequences as detailed in S3 Table. To avoid off-target effect in RNAi, we have performed off-target prediction of RNAi target sites by setting the threshold of off-target bases to 19 bp [55] and synthesized two independent RNAi constructs targeting non-overlapping regions of the receptor mRNA. dsRNA was synthesized according to the manufacturer's instructions using the MEGAscript RNAi Kit (Thermo Fisher Scientific, United States). Female flies were cold-anesthetized on ice before being injected with 0.5 µL of dsRNA at a concentration of 1,000 ng/µL into their thorax. After a period of 12 or 72 h, quantitative PCR was employed to evaluate gene knockout efficiency in ovipositor and CNS. Additionally, egg count and oviposition preference were quantified in flies with knockdown of the relevant genes. As a control, dsRNA targeting the GFP gene was injected into the thorax of female flies.

## Grik2b and Grik2c sequences analysis

The Grik2b and Grik2c sequences were analyzed for similarity to proteins of other insects using the BLAST tool (https://www.ncbi.nlm.nih.gov/). Multiple sequence alignment of the Grik2b and Grik2c amino acid sequences was conducted with Muscle using default parameters, and the alignment results were visualized with TexShade. Phylogenetic analysis of the sequences was performed using MEGA11, and maximum likelihood (ML) tree reconstruction was carried out using the Poisson model with uniform rates. The ML heuristic search utilized the nearest neighbor-change method, and the initial tree was selected by applying the neighbor joining method to a matrix of pairwise distances estimated using the JTT method. The accuracy of the tree was assessed through bootstrapping with 100 replicates.

## Molecular docking

The three-dimensional structures of Grik2b and Grik2c were modeled using Swiss-Model (https://swissmodel.expasy.org/). The ProCheck program in SAVES V7.0 (https://saves.mbi.ucla.edu/) was utilized to assess the quality and rationality of the protein models. Molecular docking of ligands and receptors was conducted with AutoDock Vina 1.1.2, and the resulting docking outcomes were imported into PyMOL for further analysis and visualization.

## Proteolysis protection assays

The ligand-binding domains (LBDs) of Grik2b and Grik2c were cloned into pET SUMO expression vectors for expression in *Escherichia coli*. The recombinant protein in the precipitates was collected and renatured using a Ni-NTA affinity column (Invitrogen) following the manufacturer's protocols. After incubation with the indicated ligands (Glutamate, D-AP5, and NBQX) for 30 min at 37 °C, the LBDs of Grik2b and Grik2c were digested by trypsin at a 1:20 ratio for half an h. Proteolysis results were determined by western blot analysis. The digested proteins were separated on a 12% SDS-PAGE gel and transferred to a hydrophilic polyvinylidene fluoride imprinted membrane with a pore size of 0.45 µm. Subsequently, the membrane was blocked in 5% skim milk in PBST for 1.5 h at room temperature. The primary antibody for 6 × His Tag was applied to incubate with membranes overnight at 4°C, followed by incubation with goat anti-mouse IgG as the secondary

antibody for 1 h at room temperature. Finally, after extensive washing in PBST, visualization was achieved using High-sensitive Plus ECL luminescence reagent (Sangon Biotech) on an Amersham Imager 600 (GE Healthcare, United States).

## Binding ability of Grik2b and Grik2c LBDs to glutamate by SPR assays and MST assays

Firstly, binding assays (SPR) were conducted using a Biacore T200 instrument (BIAcore, Cytiva, Sweden) in a running buffer consisting of 1.37M NaCl, 26.8 mM KCL, 1 mM $Na_2HPO_4$, and 17.6 mM $KH_2PO_4$. However, only the content of Grik2c protein fulfills the experimental requirements, whereas the content of Grik2b protein is excessively low to carry out the SPR experiment. Thus, the LBD of the Grik2c was immobilized as a solid phase on CM5 chips through amine coupling. Protein fixation parameters were set at a concentration of 30–50 μg/ml, pH 4.0, flow rate of 10 μl/min, and temperature of 25 °C. Kinetics analysis was performed using the Biacore T200 software with the application of the 1:1 Langmuir binding model.

Since the concentration of the purified Grik2b protein was too low, we utilized the LBD of Grik2b to assess its interaction with specific ligands (Glutamate, D-AP5, and NBQX) with the MST experiments, which requires a smaller amount of protein and is capable of assessing the binding ability of receptors and ligands just like SPR. The NT-647-NHS fluorescent dye (NanoTemper, München, Germany) was covalently linked to the target protein. The compounds (e.g., glutamate) were diluted serially (ranging from 61 nM to 2 mM) using assay buffer and then combined with Grik2b protein at a 1:1 volume ratio. The experimental buffer consisted of 100 mM Tris-HCl (pH 7.8), 1.5 M NaCl, and 0.5% Tween 20. Following a 15-minute incubation at room temperature, the mixed sample was loaded into the Monolith NT.115 instrument (NanoTemper, München, Germany) using premium-treated capillaries. Statistical analysis was conducted using NanoTemper analysis software for data interpretation and presentation.

## Grik2b and Grik2c expression in *Xenopus* oocytes and functional analysis

Two-electrode voltage clamp recordings were performed according to the methods and protocols of Han and colleagues [56]. The coding sequences of the Grik2b and Grik2c genes were amplified from cDNA and subsequently cloned into pT7Ts vectors for expression in *Xenopus* oocytes. The two plasmids were linearized with SalI and XbaI, respectively, followed by further purification using a universal DNA purification kit (Tiangen Biotech). The target gene's cRNA was synthesized using the HiScribe T7 High Yield RNA Synthesis Kit (NEB) and then diluted to a concentration of 1 μg/μL with RNase-free water. Subsequently, 27.6 nL of cRNA was injected into mature oocytes which were incubated at 18 °C in a bath solution composed of 100 mM KCl, 1 mM NaCl, and 5 mM HEPES (pH 7.5), supplemented with $MgCl_2$, $CaCl_2$, and $BaCl_2$. After incubation for 2 days in the Bath solution, two-electrode voltage clamp recordings were made using a 3M KCl agarose electrode. Specifically, the voltage step protocol was configured as maintaining the potential at 0 mV for 10 ms, followed by a −60 mV step for 20 ms, and then reverting to 0 mV for 20 ms. The stimulation was repeated at 1 s intervals during the recording process. Concanavalin A (0.6 mg/mL) (impaled to block desensitization) and glutamate (3 mM) were perfused into the oocytes continuously. The oocytes not injected with cRNA were used as the control. AxoClamp 900A was used to detect the current generated by glutamate stimulation, and the current signal was digitized using Digidata 1440A and pClamp 10. The disparity in current values recorded at −60 and 0 mV is designated as the current value corresponding to the glutamate stimulation of the cell. The current values were counted using Clamfit software, normalized and plotted using GraphPad 10.0 software. Glutamate and Con A were purchased from Sangon-Biotech while other buffer solutions were obtained from Sigma–Aldrich.

## Drug treatments

Prior to application, all drugs (D-AP5 and NBQX) were pre-diluted in DMSO to create a stock solution. Subsequently, the drug was prepared in sterile water to form a working solution of the appropriate concentration (100 μM). Gravid Female flies were cold-anesthetized by placing them on ice, after which 0.5 μL of the drug was microinjected into their thorax. The number of eggs laid and oviposition preference for gut bacteria added fruit were assessed 72 h later.

## Female movement analysis

To test the influence of gene knockdown on the motion of females, Ethovision XT 11.5 (Noldus Information Technologies, the Netherlands), specifically designed for studying the complex behaviors of animals, was employed to analyze the movement of the females. Specifically, gravid females were cold-anesthetized on ice before being injected with 0.5 μL of dsRNA at a concentration of 1,000 ng/μL into their thorax. After 72 h, one female was transferred into the experimental area (a 3.5 cm diameter petri dish). Twelve individuals were tested for each treatment. The movement of the female was recorded using a DV (HDR-XR500, Japan) for 30 min. Subsequently, the move distance, move speed, and activity of the females were analyzed with Ethovision XT 11.5. The static silhouette method was selected as the detection method, and the grayscale area range was set from 49 to 200, with frame dropping correction and trajectory smoothing enabled. The default configuration of the software was utilized to draw motion trajectories and the motion heat map.

## Data analysis

The statistical methods used in the study are indicated in the figure legends. Differences were considered significant when $P < 0.05$. All the data were analyzed using GraphPad Prism version 10 (GraphPad Software, La Jolla, California, USA; www.graphpad.com).

## Supporting information

**S1 Fig. Differential gene expression in the ovipositor of gravid females compared to that of younger females.** **(A)** PCA reveals variations in gene expression across different developmental stages of female ovipositors. **(B)** Ovipositor differentially expressed genes between 12-day old females and newly emerged females (0-day old). **(C)** Ovipositor differentially expressed genes between 12-day old females and 3-day old females. **(D)** Ovipositor differentially expressed genes between 12-day old females and 6-day old females. **(E)** Ovipositor differentially expressed genes between 12-day old females and 9-day old females.
(DOCX)

**S2 Fig. Expression trend of ovipositor genes at different development stages.**
(DOCX)

**S3 Fig. Similarity of the ligand-binding domains of Grik2b to those found in other insects.**
(DOCX)

**S4 Fig. Similarity of the ligand-binding domains of Grik2c to those found in other insects.**
(DOCX)

**S5 Fig. Microscopic Observation of the Internal Structure of the Ovipositor. (A)** Hematoxylin and Eosin (HE) staining showed the muscle cells were enveloped by neurons in ovipositor. **(B)** Muscle and neuron fibers labeled by Phalloidin (red) and Horseradish Peroxidase (HRP) (green), respectively, in the ovipositor. **(C)** Muscle and neuron cells labeled by Phalloidin (red) and HRP (green), respectively, in the ovipositor. The nucleus is stained with DAPI (blue). **(D)** NMJ observed by transmission electron microscopy in the ovipositor. **(E)** No Grik2b and Grik2c signal (red) identified in NMJs of the ovipositor by FISH with sense probes. **(F)** Weak Grik2b and Grik2c signal identified in NMJs of the ovipositor by FISH after RNase treatment.
(DOCX)

**S6 Fig. A device used for testing oviposition preference.**
(DOCX)

**S7 Fig. Influence of Grik2a, Grik2b, or Grik2c knockdown on oviposition. (A)** Expression of Grik2b in ovipositor 72 h after Grik2b dsRNA injection ($n = 5$, $P = 0.0237$, Independent sample student $t$ test). **(B)** Expression of Grik2c in ovipositor 72 h after Grik2b dsRNA injection ($n = 5$, $P = 0.8521$, Independent sample student $t$ test). **(C)** Expression of Grik2c in ovipositor 72 h after Grik2c dsRNA injection ($n = 5$, $P = 0.0211$, Independent sample student $t$ test). **(D)** Expression of Grik2b in ovipositor 72 h after Grik2c dsRNA injection ($n = 5$, $P = 0.9433$, Independent sample student $t$ test). **(E)** Eggs in ovary of female after Grik2b/c being knockdown ($n = 13$, $F_{(2,36)} = 0.6926$, $P = 0.5068$, Ordinary one-way ANOVA). **(F)** Expression of Grik2b in ovipositor 72 h after additional multiple non-overlapping RNAi constructs injection ($n = 5$, $P = 0.0002$, Independent sample student $t$ test). **(G)** Expression of Grik2c in ovipositor 72 h after additional multiple non-overlapping RNAi constructs injection ($n = 5$, $P = 0.0002$, Independent sample student $t$ test). **(H)** Total eggs laid by gravid females injected with additional multiple non-overlapping RNAi constructs ($n = 15$, $F_{(2,84)} = 0.7331$, $P < 0.0001$, Two-way ANOVA). **(I)** Oviposition preference of females injected with additional multiple non-overlapping RNAi constructs (dsGFP: $n = 16$, $P < 0.0001$; dsGrik2b: $n = 16$, $P = 0.2401$; dsGrik2c: i = 16, $P = 0.969$; Paired sample student $t$ test). **(J)** Expression of Grik2a in ovipositor 72 h after Grik2a dsRNA injection ($n = 4$, $P = 0.0009$, Independent sample student $t$ test). **(K)** Total eggs laid by gravid females injected with Grik2a RNAi constructs (n = 15, $F_{(1,56)} = 3.373$, $P = 0.0583$, Two-way ANOVA). **(L)** Oviposition preference of females injected with with Grik2a RNAi constructs. (dsGFP: $n = 16$, $P < 0.0001$; dsGrik2a: $n = 16$, $P = 0.0079$; Paired sample student $t$ test). The data underlying this figure can be found in S6 Data.
(DOCX)

**S8 Fig. Motion ability of the females with the KAR and glutamate synthesis genes knocked down. (A)** Example of female movement trajectories and heat maps. **(B)** Move distance comparison between KAR, GluDH, GluSN knocked down females and the control ($n = 12$, $F_{(4,55)} = 0.3695$, $P = 0.8294$, One-way ANOVA). **(C)** Move speed comparison between KAR, GluDH, GluSN knocked down females and the control ($n = 12$, $F_{(4,55)} = 0.3847$, $P = 0.8187$, One-way ANOVA). **(D)** Manic time comparison between KAR, GluDH, GluSN knocked down females and the control ($n = 12$, $F_{(4,55)} = 0.4673$, $P = 0.7595$, One-way ANOVA). **(E)** Manic frequency comparison between KAR, GluDH, GluSN knocked down females and the control ($n = 12$, $F_{(4,55)} = 0.5589$, $P = 0.6934$, One-way ANOVA). **(F)** Active time comparison between KAR, GluDH, GluSN knocked down females and the control ($n = 12$, $F_{(4,55)} = 0.2834$, $P = 0.8875$, One-way ANOVA). **(G)** Active frequence comparison between KAR, GluDH, GluSN knocked down females and the control ($n = 12$, $F_{(4,55)} = 0.2707$, $P = 0.8956$, One-way ANOVA). **(H)** Still time comparison between KAR, GluDH, GluSN knocked down females and the control ($n = 12$, $F_{(4,55)} = 0.713$, $P = 0.5866$, One-way ANOVA). **(I)** Still frequence comparison between KAR, GluDH, GluSN knocked down females and the control ($n = 12$, $F_{(4,55)} = 0.2853$, $P = 0.8863$, One-way ANOVA). The data underlying this figure can be found in S7 Data.
(DOCX)

**S9 Fig. Effects of muscle development and contraction related genes on oviposition and Gria2b/c expression. (A)** Effect of knocking down muscle development and contraction genes on egg-laying ($n = 10$, $F_{(4,45)} = 16.5$, $P < 0.001$, One-way ANOVA). **(B–E)** Effect of Grik2b/c knockdown on the expression of muscle development and contraction genes (**B**: $n = 3$, $F_{(2,6)} = 0.1413$, $P = 0.871$; **C**: $n = 3$, $F_{(2,3)} = 0.7379$, $P = 0.517$; **D**: $n = 3$, $F_{(2,6)} = 0.0618$, $P = 0.9406$; **E**: $n = 3$, $F_{(2,6)} = 0.0156$, $P = 0.9845$; One-way ANOVA). The data underlying this figure can be found in S8 Data.
(DOCX)

**S10 Fig. Construction and evaluation of LBDs of Grik2b and Grik2c model. (A)** Schematic of Grik2b LBD model. **(B)** The quality of the Grik2b LBD model assessed by Ramachandran Plot. **(C)** Schematic of Grik2c LBD model. **(D)** The quality of the Grik2c LBD model assessed by Ramachandran Plot.
(DOCX)

**S11 Fig. RNAi efficiency of the glutamate synthesis genes. (A)** RNAi efficiency of GluDH ($n = 4$, $P = 0.0047$, Independent sample student $t$ test). **(B)** RNAi efficiency of GluSN ($n = 5$, $P = 0.0321$, Independent sample student $t$ test). The data underlying this figure can be found in S9 Data.
(DOCX)

**S12 Fig. The effect of thorax or abdomen injection of dsRNA on the expression of KARs and oviposition. (A)** Grik2b and Grik2c expression in CNS 12 h after thorax injection of dsRNA targeting Grik2b or Grik2c (Grik2b: $n = 4$, $P = 0.031$; Grik2c: $n = 4$, $P = 0.0042$; Independent sample student $t$ test). **(B)** Grik2b and Grik2c expression in ovipositor 12 h after thorax injection of dsRNA targeting Grik2b or Grik2c (Grik2b: $n = 5$, $P = 0.8635$; Grik2c: $n = 5$, $P = 0.9123$; Independent sample student $t$ test). **(C)** Grik2b and Grik2c expression in ovipositor 24 h after thorax injection of dsRNA targeting Grik2b or Grik2c (Grik2b: $n = 4$, $P < 0.001$; Grik2c: $n = 4$, $P = 0.0216$; Independent sample student $t$ test). **(D)** Grik2b and Grik2c expression in CNS 24 h after thorax injection of dsRNA targeting Grik2b or Grik2c (Grik2b: $n = 4$, $P = 0.009$; Grik2c: $n = 4$, $P = 0.0144$; Independent sample student $t$ test). **(E)** Oviposition preference to the gut bacteria strain added fruit of females with dsRNA of Grik2b and Grik2c injected into thorax for 24 h (dsGFP: $n = 10$, $P < 0.0001$; dsGrik2b: $n = 10$, $P = 0.9543$; dsGrik2c: $n = 10$, $P = 0.7336$; Paired sample student $t$ test). **(F)** Total eggs laid by females with dsRNA of Grik2b and Grik2c injected into thorax for 24 h ($n = 10$, $F_{(2,54)} = 5.945$, $P = 0.0046$, Two-way ANOVA). **(G)** Grik2b and Grik2c expression in ovipositor 6 h after abdomen injection of dsRNA targeting Grik2b or Grik2c (Grik2b: $n = 4$, $P < 0.001$; Grik2c: $n = 4$, $P = 0.002$; Independent sample student $t$ test). **(H)** Grik2b and Grik2c expression in CNS 6 h after abdomen injection of dsRNA targeting Grik2b or Grik2c (Grik2b: $n = 4$, $P = 0.6978$; Grik2c: $n = 4$, $P = 0.2639$; Independent sample student $t$ test). **(I)** Grik2b and Grik2c expression in ovipositor 12 h after abdomen injection of dsRNA targeting Grik2b or Grik2c (Grik2b: $n = 4$, $P = 0.0041$; Grik2c: $n = 4$, $P = 0.001$; Independent sample student $t$ test). **(J)** Grik2b and Grik2c expression in CNS 12 h after abdomen injection of dsRNA targeting Grik2b or Grik2c (Grik2b: $n = 4$, $P < 0.001$; Grik2c: $n = 4$, $P = 0.002$; Independent sample student $t$ test). **(K)** Oviposition preference to the gut bacteria strain added fruit of females with dsRNA of Grik2b and Grik2c injected into abdomen for 12 h (dsGFP: $n = 10$, $P < 0.0001$; dsGrik2b: $n = 14$, $P = 0.8635$; dsGrik2c: $n = 15$, $P = 0.5822$; Paired sample student $t$ test). **(L)** Total eggs laid by females with dsRNA of Grik2b and Grik2c injected into abdomen for 12 h ($n = 10$ and 15, $F_{(2,70)} = 5.973$, $P = 0.004$, Two-way ANOVA). The data underlying this figure can be found in S10 Data.
(DOCX)

**S1 Table. Prediction of hydrogen bond interactions between the glutamate receptors and the ligands.**
(DOCX)

**S2 Table. Predication of binding site amino acid residue of glutamate receptors.**
(DOCX)

**S3 Table. Primer sequences used in this study.**
(DOCX)

**S1 Video. Oviposition behavior compare between Grik2b knockdown female and control (in 10 min)** . Left, Control; Right, Grik2b knockdown.
(MP4)

**S2 Video. Oviposition behavior compare between Grik2c knockdown female and control (in 10 min)** . Left: Control; Right, Grik2c knockdown.
(MP4)

**S1 Data. Source data and statistical details for** Fig 1**.**
(XLSX)

**S2 Data. Source data and statistical details for** Fig 2.
(XLSX)

**S3 Data. Source data and statistical details for** Fig 3.
(XLSX)

**S4 Data. Source data and statistical details for** Fig 4.
(XLSX)

**S5 Data. Source data and statistical details for** Fig 5.
(XLSX)

**S6 Data. Source data and statistical details for** **S7 Fig**.
(XLSX)

**S7 Data. Source data and statistical details for** **S8 Fig**.
(XLSX)

**S8 Data. Source data and statistical details for** **S9 Fig**.
(XLSX)

**S9 Data. Source data and statistical details for** **S11 Fig**.
(XLSX)

**S10 Data. Source data and statistical details for** **S12 Fig**.
(XLSX)

**S1 Raw Images. Raw images for blots.**
(TIF)

## Acknowledgments

We thank Guangwu Zhu for rearing the insects.

## Author contributions

**Conceptualization:** Bin Liu, Daifeng Cheng.

**Data curation:** Bin Liu, Daifeng Cheng.

**Formal analysis:** Daifeng Cheng.

**Funding acquisition:** Daifeng Cheng.

**Investigation:** Bin Liu, Jingwei Yang, Long Ye, Yang Xiao, Guohong Luo, Muyang He.

**Methodology:** Bin Liu.

**Supervision:** Yongyue Lu, Daifeng Cheng.

**Validation:** Daifeng Cheng.

**Visualization:** Daifeng Cheng.

**Writing – original draft:** Bin Liu, Daifeng Cheng.

**Writing – review & editing:** Guy Smagghe, Daifeng Cheng.

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
