## [Editor Report · Decision Letter 0]

25 Aug 2025

Dear Dr Cheng,

Thank you for submitting your manuscript entitled "Function redundancy of kainate receptors in regulating oviposition of oriental fruit fly" for consideration as a Research Article by PLOS Biology. Please accept our apologies for the delay in sending you an initial decision. We had wished to discuss your paper with an Academic Editor with relevant expertise, and it took us a bit longer than normal to find someone who was available to provide advice.

Your manuscript has now been evaluated by the PLOS Biology editorial staff as well as by an Academic Editor and I am writing to let you know that we would like to send your submission out for external peer review.

Once your full submission is complete, your paper will undergo a series of checks in preparation for peer review. After your manuscript has passed the checks it will be sent out for review. To provide the metadata for your submission, please Login to Editorial Manager (https://www.editorialmanager.com/pbiology) within two working days, i.e. by Aug 27 2025 11:59PM.

Kind regards,

Luke

Lucas Smith, Ph.D.

Senior Editor

PLOS Biology

lsmith@plos.org

---

## [Decision Letter · Decision Letter 1]

2 Oct 2025

Dear Dr Cheng,

Thank you for your patience while your manuscript entitled "Function redundancy of kainate receptors in regulating oviposition of oriental fruit fly" was peer-reviewed at PLOS Biology. Your manuscript has been evaluated by the PLOS Biology editors, an Academic Editor with relevant expertise, and by three independent reviewers.

The reviews are attached below. As you will see, the reviewers find the conclusions potentially interesting, however they also raise several concerns that would need to be addressed before we can consider the manuscript for publication. Reviewer 1 thinks that the data as presented doesn’t support the main conclusions and that the RNAi experiments should be strengthened, along with additional controls, better quantification of the experiments and behavioural data for Grik2a. Reviewer 2 asks for further insights into the neuronal mechanisms that controls egg-laying behaviour, an analysis of the function of the Grik2b receptor and increasing the sample size for the behavioural experiments among other points. Reviewer 3 mentions several points that would need to be clarified.

In addition, the Academic Editor has raised several points that would been to be also addressed. These include whether or not the knockdown of the receptors affect the overall vitality of the flies, as the observed phenotype (lowered oviposition, etc) could be simply an indirect consequence of that. The Academic Editor is also concerned about off-targets’ knockdown causing the phenotype.

Based on their specific comments and following discussion with the Academic Editor, it is clear that a substantial amount of work would be required to meet the criteria for publication in PLOS Biology. However, given our and the reviewer interest in your study, we would be open to inviting a comprehensive revision of the study that thoroughly addresses all the reviewers' comments. Given the extent of revision that would be needed, we cannot make a decision about publication until we have seen the revised manuscript and your response to the reviewers and Academic Editor's comments. Your revised manuscript would need to be seen by the reviewers again, but please note that we would not engage them unless their main concerns have been addressed.

We appreciate that these requests represent a great deal of extra work, and we are willing to relax our standard revision time to allow you 6 months to revise your study. Please email us (plosbiology@plos.org) if you have any questions or concerns, or envision needing a (short) extension.

**IMPORTANT - SUBMITTING YOUR REVISION**

3. Resubmission Checklist

a) *PLOS Data Policy*

b) *Published Peer Review*

d) *Blurb*

Please also provide a blurb which (if accepted) will be included in our weekly and monthly Electronic Table of Contents, sent out to readers of PLOS Biology, and may be used to promote your article in social media. The blurb should be about 30-40 words long and is subject to editorial changes. It should, without exaggeration, entice people to read your manuscript. It should not be redundant with the title and should not contain acronyms or abbreviations. For examples, view our author guidelines: https://journals.plos.org/plosbiology/s/revising-your-manuscript#loc-blurb

Sincerely,

Ines

--

Ines Alvarez-Garcia, PhD

Senior Editor

PLOS Biology

on behalf of

Lucas Smith, Ph.D.

Senior Editor

PLOS Biology

lsmith@plos.org

Reviewers' comments

Reviewer #1:

The study proposes that two kainate type iGluRs (Grik2b, Grik2c) act in both glutamatergic CNS neurons and ovipositor neuromuscular junctions (NMJs) to regulate oviposition preference and egg number in Bactrocera dorsalis. Evidence cited includes ovipositor enriched transcriptomics, FISH and histology at ovipositor NMJs, FISH colocalization with Vglut in the CNS, RNAi knockdowns affecting behavior, docking plus MST/SPR ligand binding, limited Xenopus oocyte recordings, and small molecule antagonist injections. While the hypothesis is not uninteresting, the data as presented do not yet support the central conclusions.

Major issues

1. The inference that Grik2b/c are required relies heavily on RNAi, with limited assessment of off target effects. Stronger evidence would include (i) multiple, non overlapping RNAi constructs with concordant phenotypes, (ii) rescue with RNAi resistant cDNA, and/or (iii) CRISPR loss of function plus genetic rescue. Clarify pharmacology: the antagonist used appears inconsistent with kainate receptor selectivity; specify its known target profile and justify its use here.

2. The attribution of reduced egg laying to ovipositor iGluR is correlative. Survey additional candidate loci (e.g., oviduct musculature) and, where feasible, apply tissue restricted manipulations to localize function.

3. The oviposition phenotype is described at a very coarse level. High magnification, quantitative kinematics (e.g., ovipositor extrusion frequency, duration, bout structure) would clarify if and how iGluR disruption alters the motor program.

Minor / specific issues

1. Behavioral data for Grik2a are missing. If "redundant roles" are claimed, double knockdown (Grik2b+Grik2c) should exhibit a stronger than either single (often super additive) phenotype, state and test that prediction explicitly.

2. In Fig. 2n (and 2j), both control and experimental groups lay far fewer eggs than in Fig. 2i. Such baseline drops can bias preference indices; report raw counts, normalize within day/batch, and analyze with models that account for baseline variation.

3. Include sense probe (and ideally RNase) controls and specify the exact ovipositor subregions sampled; add a schematic to anchor tissue locations.

4. Fig. 4k implies near abolition of VNC activity after knockdown, but the format is not acceptable for inference. Provide n per group, ROI definitions, ΔF/F₀ quantification, motion correction, and appropriate statistics across biological replicates.

Reviewer #2:

This paper identifies the neurotransmitter glutamate as playing a role in oviposition behavior of the oriental fly Bactrocera dorsalis. Studying oviposition behavior of this species is of interest since it is an agricultural pest affecting fruit production by laying its eggs in the fruits and very little is known on the neuronal mechanisms governing this behavior. The females use an elongated ovipositor to deposit their eggs deep inside the fruits and this organ appears to be also involved in sensing egg-laying substrates.

The authors investigate glutamate signaling after having identified by transcriptomic analysis 3 glutamate receptors (Grik2a, 2b and 2c) as being highly expressed in the ovipositors of gravid females. They focus on 2 of these receptors, Grik2b and 2c, and demonstrate via several biochemical experiments that they indeed bind to glutamate and are able to induce currents when expressed in Xenopus oocytes. They also document expression patterns of the receptors in the Central Nervous System and ovipositor by in situ hybridization.

They then demonstrate via RNAi knockdowns that Grik2b and 2c regulate oviposition behavior. They confirm these results by manipulating glutamate levels (GluDH, GluSN and vGlut RNAi knockdown) and by injecting receptor inhibitors into the flies (D-AP5, NBQX) in the same behavioral assays. Notably, the authors argue that they can distinguish two separate functions on oviposition behavior by glutamate signaling: Grik receptors expressed in the CNS appear to participate in determining egg-laying substrate preference (favoring substrates containing bacteria), while Grik receptors in the ovipositor probably directly control muscle contractions to expel the eggs since they are expressed in muscle cells (i.e.: Neuro Muscular Junction).

Overall, the data is well presented with appropriate controls. However, while this paper uses a challenging model organism and manages to perform functional manipulations (RNAi knockdowns) to demonstrate the involvement of glutamate signaling, it provides relatively little insight into the neuronal mechanisms controlling this behavior. In fact, glutamate has already been implicated in driving the egg-laying motor program in Drosophila (e.g. Cury and Axel 2023, Oliveira-Ferreira et al. 2023) but this is not mentioned in this paper. The paper seems to allude to potential applications for pest control via the use of Grik receptor inhibitors but does not explore this idea further.

Specific comments:

1. It is not clear to me why the authors didn't investigate the function of the Grik2a receptor.

2. The authors observe phenotypes when knocking down either one the Gri2b and 2c receptors throughout the paper and they mention "function redundancy of kainite receptors" in their ms title but they don't discuss the implications. How is this working? If both receptors are essential, they are not redundant…

The authors should control for specificity of their dsRNA probes, showing that dsRNA against Gri2b do not affect Grik2c levels and vice versa.

3. The sample sizes for many behavioral experiments appear quite low (e.g.: n=5 replicates in Fig 2l-m, Fig 5k-l)

4. Lines 217-218: I don't understand the authors' argument. The fact that Grik receptors respond to glutamate does not imply that they should be expressed in glutamate-producing neurons.

5. Fig 4a-d: the authors detect Grik2b/2c expression in the CNS but do not detect them by qPCR in the thorax in Fig 1h-I. Do they have an explanation?

6. Fig 4e-f: it is difficult to understand what we are looking at. The authors argue that Gri2b/2c do not colocalize with vGlut but in fact there is a bright spot in the center of the image in both cases with strong green and red signal. Is this a neuron(s)?

7. Fig 4k: It is also difficult here to understand what we are looking at. Is this the whole VNC? Only part of it?

Isn't it surprising that knocking down signaling of one neurotransmitter has such profound effects on calcium levels? Also, the images shown are only a snapshot, what are the temporal dynamics?

8. Fig 4o: the authors argue that CNS-specific knockdown of Grik2b/2c (12h after thoracic dsRNA injection) does not affect the egg-laying rate but in fact the rate is quite low, even in controls (about 100 eggs on average whereas it is usually around 200 in other experiments). Can they provide an explanation? Is it related to the shorter time of recovery after dsRNA injection?

8. Fig 5k-l: the authors inject the Grik inhibitors in the flies but this is not applicable in the field. Have they tried feeding the flies with these inhibitors?

Reviewer #3:

In the current study, the authors investigated the roles of two kainate receptors (Grik2b and Grik2c) in regulating oviposition in Bactrocera dorsalis, an agricultural insect pest, proposing that glutamatergic neurons mediate oviposition preference while ovipositor NMJs control egg-laying quantity. The authors employed multiple approaches (transcriptomics, FISH, time control RNAi, LBD, electrophysiology, behavioral assays, and pharmacological inhibition). I found this interesting manuscript to be well-written and described. The findings in this study are valuable for helping us better understand the neural conduction mechanism of the insect oviposition behaviors and have practical implications for pest control strategies. The results should be useful to many in the field.

I have a couple of extra questions about the work,

1. The author should explain whether the additional independent non-overlapping dsRNA for targets is necessary to exclude the off-target RNAi effects.

2. Even though the current study is focused on oviposition preference and egg-laying quantity, have the authors confirmed the retained eggs within the reproductive tract (such as ovaries) before/after injection to identify any effects on egg production?

---

## [Editor Report · Decision Letter 2]

18 Dec 2025

Dear Dr Cheng,

Thank you for your patience while we considered your revised manuscript "Dual Roles of Kainate Receptors in Oviposition of the oriental fruit fly" for publication as a Research Article at PLOS Biology. This revised version of your manuscript has been evaluated by the PLOS Biology editors and the Academic Editor who is satisfied by the changes made and who has commented that the revision does a very thorough job, within the constraints of their system, in meeting the reviewer requests.

Based on our Academic Editor's assessment of your revision we are likely to accept this manuscript for publication. However, before we can do so we need you to address a number of data and other policy-related requests, in a last short revision. These are detailed below.

**IMPORTANT - Please address the following editorial requests:

1) TITLE: After some discussion within the team, we would like to propose a tweak to the title to add a bit more detail and to use the scientific name of the animal studied here. If you agree, we suggest changing your title to:

"Grik2b and Grik2c kainate receptors regulate oviposition in Bactrocera dorsalis"

2) FINANCIAL DISCLOSURES STATEMENT: Please update your financial disclosures statement, in our editorial manager system, to indicate whether the sponsors or funders played any role in the study design, data collection and analysis, decision to publish, or preparation of the manuscript.

3) DATA: You may be aware of the PLOS Data Policy, which requires that all data be made available without restriction: http://journals.plos.org/plosbiology/s/data-availability. For more information, please also see this editorial: http://dx.doi.org/10.1371/journal.pbio.1001797

>>Under this policy, we ask that you please provide all the raw RNA-seq data, generated here, on a publicly available data repository.

>>Please ensure that your Data Statement in the submission system accurately describes where your data can be found.

4) DATA: In addition to providing the raw RNA-seq data, we also request that you provide the underlying data for the rest of your figures. Note, for the other (non-RNA-seq) studies presented in your manuscript we do not require all raw data. Rather, we ask that all individual quantitative observations that underlie the data summarized in the figures and results of your paper be made available in one of the following forms:

a. Supplementary files (e.g., excel). Please ensure that all data files are uploaded as 'Supporting Information' and are invariably referred to (in the manuscript, figure legends, and the Description field when uploading your files) using the following format verbatim: S1 Data, S2 Data, etc. Multiple panels of a single or even several figures can be included as multiple sheets in one excel file that is saved using exactly the following convention: S1_Data.xlsx (using an underscore).

b. Deposition in a publicly available repository. Please also provide the accession code or a reviewer link so that we may view your data before publication.

>>Regardless of the method selected, please ensure that you provide the individual numerical values that underlie the summary data displayed in the following figure panels as they are essential for readers to assess your analysis and to reproduce it:

Fig 2C-J; Fig 3E-M; Fig 4G-J,L-P; Fig 5 G-L;

Fig S7A-L; Fig S8-I; Fig S9A-E; Fig S11A-B; Fig S12A-L

>>Please also ensure that figure legends in your manuscript include information on where the underlying data can be found, and ensure your supplemental data file/s has a legend.

>>Please ensure that your Data Statement in the submission system accurately describes where your data can be found.

5) BLOTS AND GELS: We require the original, uncropped and minimally adjusted images supporting all blot and gel results reported in an article's figures or Supporting Information files. We will require these files before a manuscript can be accepted so please prepare and upload them now. Please carefully read our guidelines for how to prepare and upload this data: https://journals.plos.org/plosbiology/s/figures#loc-blot-and-gel-reporting-requirements

6) CODE: Per journal policy, if you have generated any custom code during the course of this investigation, please make it available without restrictions. Please ensure that the code is sufficiently well documented and reusable, and that your Data Statement in the Editorial Manager submission system accurately describes where your code can be found.

We expect to receive your revised manuscript within three weeks.

*Published Peer Review History*

*Press*

Sincerely,

Luke

Lucas Smith, Ph.D.

Senior Editor

lsmith@plos.org

PLOS Biology

---

## [Editor Report · Decision Letter 3]

7 Jan 2026

Dear Dr Cheng,

Thank you for the submission of your revised Research Article "Grik2b and Grik2c kainate receptors regulate oviposition in Bactrocera dorsalis" for publication in PLOS Biology and thank you also for addressing our last editorial requests in this revision. On behalf of my colleagues and the Academic Editor, Mariana Wolfner , I am pleased to say that we can in principle accept your manuscript for publication, provided you address any remaining formatting and reporting issues. These will be detailed in an email you should receive within 2-3 business days from our colleagues in the journal operations team; no action is required from you until then. Please note that we will not be able to formally accept your manuscript and schedule it for publication until you have completed any requested changes.

**IMPORTANT: As discussed over email, I have updated your submission to include an updated 'Supplementary figures and tables file', and 'Raw images' file. Please do take a moment to double check that everything looks good after this change.

PRESS

Sincerely,

Lucas Smith, Ph.D.

Senior Editor

PLOS Biology

lsmith@plos.org